J Physiol 603.14 (2025) pp 4063–4090

4063

# Out of the single-neuron straitjacket: Neurons within assemblies change selectivity and their reconfiguration underlies dynamic coding

Fabrizio Londei[1,2] , Francesco Ceccarelli[1,3], Giulia Arena[1,2,3], Lorenzo Ferrucci[1], Eleonora Russo[4], Emiliano Brunamonti[1] and Aldo Genovesio[3,5]

[1] Department of Physiology and Pharmacology, Sapienza University of Rome, Rome, Italy
[2] PhD program in Behavioral Neuroscience, Sapienza University of Rome, Rome, Italy
[3] Institute of Biochemistry and Cell Biology (IBBC), National Research Council of Italy (CNR), Monterotondo Scalo, Roma, Italy
[4] The BioRobotics Institute, Department of Excellence in Robotics and AI, Scuola Superiore Sant'Anna, Pisa, Italy
[5] Department of Pharmaceutical Sciences, University of Eastern Piedmont Amedeo Avogadro, Novara, Italy

Handling Editors: Richard Carson & Madeleine Lowery

The peer review history is available in the Supporting Information section of this article (https://doi.org/10.1113/JP288015#support-information-section).

The Journal of Physiology

**Graphical abstract legend** Traditionally, single neurons coding activity has been evaluated using all the spikes recorded. We have shown that even neurons not encoding any task variable when considering all their spikes, defined as full-spikes activity (FSA), can display emergent coding activity when considering only spikes fired in co-ordination, defined as assembly-spikes activity (ASA), with other neurons of the same assembly. A virtual neuron is shown, as created for illustrative purposes, recorded in a two-stimuli discrimination task that required choosing one of the two stimuli displayed on the right and left in each trial, based on which of the two stimuli presented sequentially earlier was either farther from the centre (distance task) or had a longer duration (duration task). Two task variables are shown on the right

This article was first published as a preprint. Londei F, Ceccarelli F, Arena G, Ferrucci L, Russo E, Brunamonti E, Genovesio A. 2024. Out of the single-neuron straitjacket: neurons within assemblies change selectivity and their reconfiguration underlies dynamic coding. bioRxiv. https://doi.org/10.1101/2024.10.03.616400

Code availability: CADopti code is available at https://github.com/DurstewitzLab/CADopti.

and left, comprising the response direction and the goal colour chosen in example trials. The middle panel shows that this neuron does not display any coding activity when considering only its FSA (grey spikes). However, the lower panel, where trials are sorted either by response direction or by the goal colour, shows that, considering the ASA (coloured spikes), the neuron contributes to coding both goal and response, which was otherwise undetectable at the level of the FSA. Specifically, the neuron shows higher activity for the left response when it co-ordinates with Assembly 1, and for the red goal when it co-ordinates with Assembly 2.

**Abstract**   We investigated cell assemblies in the frontal cortex of macaques during two discrimination tasks. Focusing on the period of goal-action transformation, we extracted spikes fired during assembly activation from the full neural activity and showed that the contribution of a neuron to assembly coding, when it co-ordinates with other assembly neurons, differs from its coding in isolation. Neurons, with their flexible participation to multiple assemblies, contributed to the encoding of new information not encoded by the neurons alone. Even non-discriminative neurons acquired selectivity as part of the collective activity of the assemblies. Thus, neurons in their assemblies process distinct information for various purposes as a chess simul master, playing on multiple chessboards. The reconfiguration of the participation of the neurons into different assemblies in the goal-action transformation process translated into a dynamic form of coding, whereas minimal reconfiguration was associated with the static goal coding of the memory period.

(Received 31 October 2024; accepted after revision 11 June 2025; first published online 9 July 2025)

**Corresponding author** A. Genovesio: Department of Pharmaceutical Sciences, University of Eastern Piedmont Amedeo Avogadro, 28100 Novara, Italy.    Email: aldo.genovesio@uniupo.it

### Key points

- Traditionally, the coding properties of a neuron are studied using all its activity (full-spikes), irrespective of its co-ordination with different groups of neurons.
- With an assembly centered approach, we can determine the neuron's coding properties not in absolute terms, but relative to the assembly of neurons with which it co-ordinates.
- When neurons are studied in different assemblies—focusing only on the spikes fired during assembly coordination (assembly-spikes)—they can contribute to the coding of different variables.
- The coding flexibility of the same neuron in multiple assemblies increases the amount of information it can contribute to encoding compared to isolated neurons.
- Dynamic coding, as opposed to static coding, as observed during the goal-action transformation process, can be explained by an increase in the reconfiguration of active assemblies, with neurons contributing to the coding of different variables in different epochs, depending on which assembly is active.

## Introduction

Hebb's theoretical framework posits that perception and cognition are underpinned by the activity of tightly interconnected neural networks, termed cell assemblies (Hebb, 1949). Cell assemblies are defined as a subset of neurons with significant coactivation behaviour and have been proposed to serve as the fundamental building blocks for information processing in the brain (Buzsáki, 2010; Harris, 2005; Nicolelis et al., 1997). Within this

**Fabrizio Londei** earned a master's degree in Applied Mathematics from Sapienza University of Rome (Italy), where he also completed his PhD in Behavioral Neuroscience. His research focuses on data analysis and computational neuroscience, particularly investigating the role of cell assemblies at multiple levels, from coding to functional connectivity, and their relationship with dynamic and static coding schemes of information in memory. He is currently a postdoctoral researcher at the Instituto de Neurociencias CSIC-UMH (Spain).

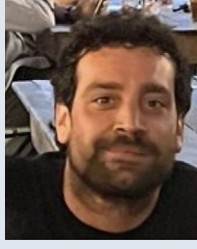

framework, neuronal assemblies may include neurons widely dispersed across different brain regions, forming intricate functional circuits through direct or indirect connections or shared input.

It is important to acknowledge that the terms 'assembly' or 'ensemble' have been used with varying definitions in many neurophysiological studies. For example, in studies examining the coding ability of neurons in a brain area, neurons are frequently grouped together, but not based on their co-ordinated activity. Panzeri et al. (2022) distinguish two types of population codes: the *independent population codes* that rely on measuring the information available in the joint representation of single neurons neglecting the functional interaction between neurons (Backen et al., 2018; Bernardi et al., 2020; Fascianelli et al., 2024; Wilson & McNaughton, 1993), and the *structured correlated population codes*, intended as the codes for which functions depend on the structure of correlations generated by the co-ordinated activity of neurons.

Co-ordinated firing between neurons reflects the organization of functional connectivity between individual cells within their assemblies. These neuron assemblies are not rigid and can change dynamically in size and composition (Sakurai, 1999). Recent studies in non-human primates have primarily used cross-correlation analysis between pairs of neurons to investigate co-ordinated activity in cell assemblies (Constantinidis et al., 2001; Funahashi & Inoue, 2000; Mione et al., 2019; Nougaret & Genovesio, 2018; Riehle et al., 1997; Sakamoto et al., 2008; Sakurai & Takahashi, 2006; Tsujimoto et al., 2008). In the present study, we used an assembly detection method proposed by Russo and Durstewitz (2017) to investigate co-ordinated activity within small groups of simultaneously recorded neurons.

Previous studies investigating the temporal evolution in co-ordination patterns between neurons, revealed that brain regions can alter their functional connectivity both during task execution (Cole et al., 2013) and during rest, demonstrating a high degree of flexibility (Yin & Kaiser, 2021). Our previous research (Marcos et al., 2019), exposed the dynamic reconfiguration of the prefrontal processing of the neural network responsible for encoding goals during goal-action transformation (i.e. when a goal kept in memory is transformed into the appropriate action for its acquisition). Building upon this observation, the present study investigates whether the neural flexibility associated with dynamic coding reflects the flexible participation of neurons in multiple assemblies that are active at different times.

An important question in the study of cell assemblies concerns the advantages deriving from assembly co-ordinated activity. Previous studies have demonstrated that an increase in correlation between neurons can lead to improved readout by downstream brain areas (Panzeri et al., 2022). This benefit, however, may come at the cost of reducing the information encoded by the neuron (Abbott & Dayan, 1999; Zohary et al., 1994), but, at the same time, it has been proposed to possibly increase the number of variables encoded within the population (Silveira & Ii, 2014). In terms of advantages, when the activity of a neuronal population becomes more correlated, the number of spikes arriving within the integration window of a downstream neuron increases, eventually pushing it towards its firing threshold and leading to a higher frequency of discharge. In support of this hypothesis, Valente et al. (2021) found that an increase in neural correlation was associated with improved task performance. Additionally, Gill et al. (2020) demonstrated that a synchronous two-photon optogenetic stimulation of less than 20 neurons in the mouse's olfactory bulb could be detected by mice, highlighting the importance of correlated activity in neural processing.

Our study aims to investigate, in a distance and duration discrimination task (Benozzo et al., 2023; Genovesio et al., 2009, 2011, 2012), how the co-ordination of neurons in assemblies and specifically their flexibility in participating in different assemblies may support the downstream propagation of information. It is well established that neurons can engage in different computations by multiplexing different signals with their activity (Caruso et al., 2018; Jun et al., 2022; Kremer et al., 2020). Although single neurons constitute the basic units of the nervous system, their impact on information processing is contingent on their interaction with the specific synaptic connection patterns of the underlying neural circuits (Luo, 2021). Therefore, it becomes clear that the assembly, rather than the neuron alone, should be considered as the substrate for information processing. In the present study, we investigated whether participation in different assemblies can further enhance the neuron's contribution to information propagation and reveal coding properties even of neurons otherwise deemed non-discriminative when analysed in isolation. As a helpful simplification, we use the term 'contribute to coding' to refer to the neuron's contribution to information coding within an assembly. When discussing the coding properties of a neuron within a given assembly, we specifically refer to its role in participation in the assembly coding. This involves considering its impact, in conjunction with other assembly members, on the downstream neurons. A further objective of the present study is to test whether a highly dynamic representation of information, such as goal coding in the goal-action transformation process, reflects a reconfiguration of the active assemblies, as proposed by Marcos et al. (2019). Although a dynamic coding scheme has been described in past studies (Ceccarelli et al., 2023; Mendoza-Halliday & Martinez-Trujillo, 2017; Meyers, 2018; Meyers et al.,

2008, 2012), its relationship with assembly neuronal reconfiguration has yet to be explored.

We demonstrate that neurons have distinct coding properties when they co-ordinate their activity within different assemblies, contributing to the encoding of a new variable, or even the same variable but with a different neural code. As a result, the same neuron participated in coding multiple types of information, which we refer to as multiple selectivity. We also found that a high level of reconfiguration of the neuronal coding properties in different assemblies explains the dynamic forms of coding.

## Methods

### Ethical approval

The present study is a data analysis study based on experimental recordings made previously in cited published works. At the time the recordings were obtained, all animal surgical and experimental procedures were approved in advance by the National Institute of Mental Health Animal Care and Use Committee [LSN_03_05] and followed the National Institutes of Health Guide for the Care and Use of Laboratory Animals (1996) and were approved by the National Institute of Mental Health Animal Care and Use Committee. The authors understand the ethical principles under which *The Journal of Physiology* operates and confirm that this work complies with its animal ethics checklist.

### Animals

Two adult male rhesus monkeys (*Macaca mulatta*) weighing 8.5 and 8.0 kg were used in this study. The monkeys were trained prior to surgery and the start of recordings for a period of ∼2 years, on a Monday to Friday schedule with two resting days during the weekend. To motivate the animals during training and neural recordings, water intake was controlled. Monkeys had full access to dry food. After the daily experimental sessions, monkeys received fresh food such as fruit and vegetables. Weight was monitored several times a week and maintained above 85% of the weight of the monkey before starting the water control schedule. The animals' weight and general health conditions were carefully monitored by full-time on-site veterinary staff. The monkeys were paired housed unless adverse outcomes precluded temporarily the pairing. The information about the care and welfare of the animals is as reported in previously published articles that used the same task/animals (Benozzo et al., 2021, 2023; Genovesio et al., 2009, 2011, 2012, 2015; Marcos et al., 2017).

### Tasks

Monkeys were trained to perform distance and duration discrimination tasks. Figure 1 shows the task events of the duration (Fig. 1*A*) and the distance (Fig. 1*B*) discrimination tasks. The monkeys were presented with two stimuli sequentially on a screen, and they had to determine which stimulus was farther from a reference point or longer in duration. To interface with the tasks, three infrared switches were placed in front of the monkeys. The trials in both tasks followed the same temporal structure. However, the two stimuli had different durations in the duration discrimination task. Each stimulus was either a blue circle with a 3° diameter or a 3° × 3° red square. If the first stimulus (S1) was the blue circle, the second stimulus (S2) was the red square and vice versa.

Each trial started when the monkeys pressed the central switch. A central stimulus was then displayed for either 400 or 800 ms, followed by the onset of S1. In the distance task, S1 was presented for 1000 ms and was positioned at a distance of 8–48 mm (in increments of 8 mm) above or below the reference point. In the duration task, S1 was positioned at a fixed distance of 24 mm from the reference point, but its duration varied between 200 and 1200 ms in increments of 200 ms. After 400 or 800 ms, and in a minority of recordings also 1600 ms, S2 was shown for a fixed duration of 1000 ms in the distance task and a variable duration ranging from 200 to 1200 ms increments of in 200 ms in the duration task, and never equal to the duration of S1. In the distance task, the distance of S2 was selected from the same range as S1 distances, but it was always different from S1's distance. In the duration task, S2 was always presented at the same distance (24 mm) from the reference point as S1. Figure 1*C* shows a representation of the possible pairs of durations used, with respect to the duration task, or possible pairs of distances used, with respect to the distance task, for the two presented stimuli. The recordings were obtained from the prefrontal cortex (Fig. 1*D*).

A delay of 0, 400 or 800 ms (D2) followed the disappearance of S2 and preceded the reappearance of the two stimuli, which served as a go-signal. Each stimulus was presented 40 mm to the right or left of the central stimulus, randomly selected. The monkeys had 6 s to select the stimulus (red or blue goal) that was farther from the reference point in the distance task or had lasted longer in the duration task by touching the corresponding switch. Before the go-signal, the monkeys could not plan any motor response. Correct choices were rewarded with 0.1 mL of fluid, whereas acoustic feedback followed incorrect choices. All the task variables, including the duration of D1 and D2 and the colour and shape of the stimuli, were determined pseudorandomly. Figure 1*E* shows a schematic of the task epochs used for the main

analyses. The tasks were presented in a block design, such that no cues were necessary nor provided during the tasks to indicate task type. More information about the two experimental tasks is provided in Genovesio et al. (2009) (2011) (2012) and (2015).

## Surgery

Chambers were placed surgically over the exposed dura mater of the left frontal lobe along with head restraint devices. The procedure followed aseptic techniques and utilized isoflurane anaesthesia (1–3% to effect). Monkey

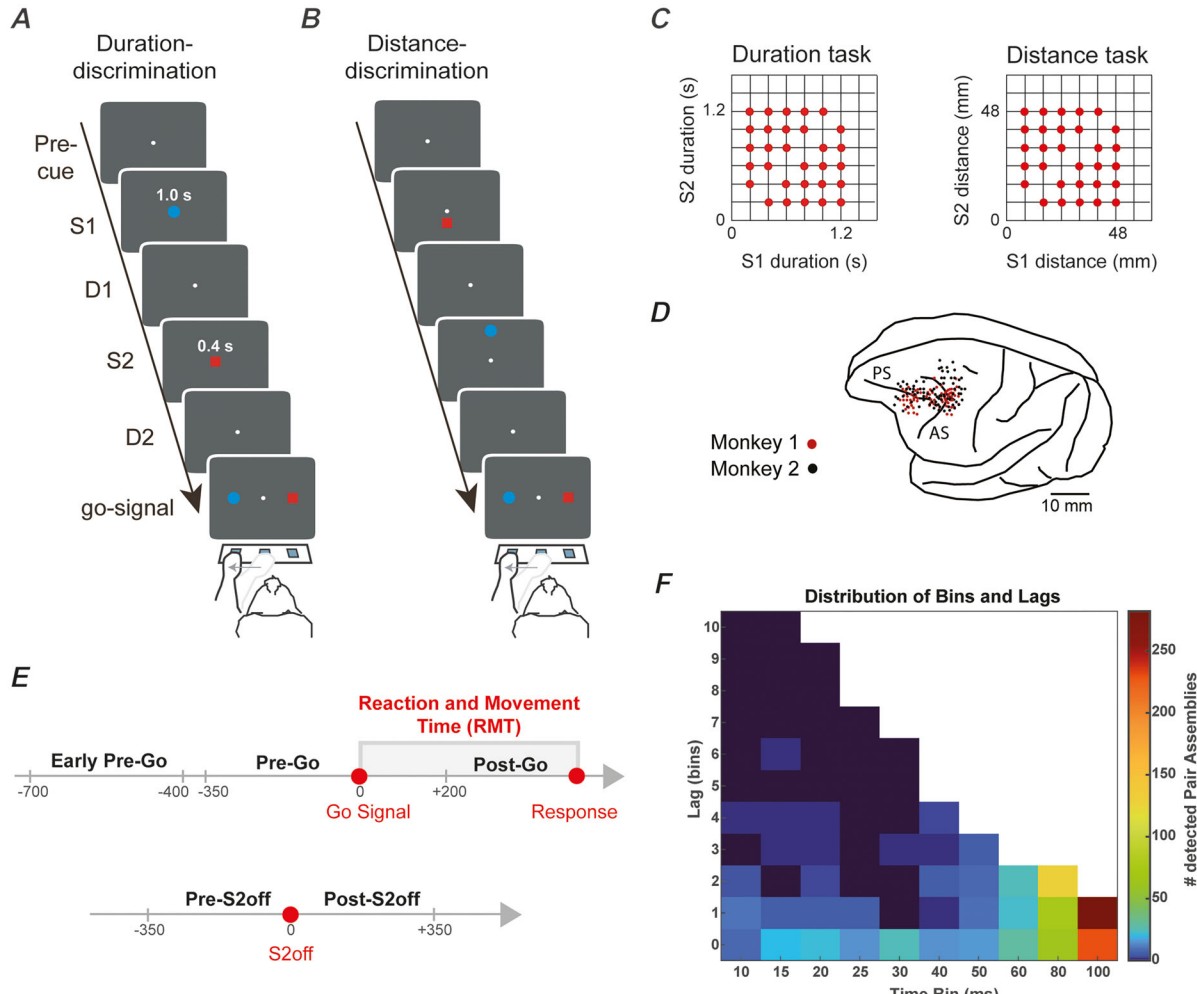

**Figure 1. Experimental tasks, stimuli distributions and recording locations**
*A* and *B*, temporal sequence of the task events during duration (*A*) and distance (*B*) discrimination tasks. In both tasks, two stimuli were sequentially presented on a screen, separated by a delay and followed, after a second delay, by the reappearance of the two stimuli that served as targets. Monkeys were required to select which of the two stimuli lasted longer, in the duration task, or was presented farther, in the distance task. After their choice, a reward was delivered in the correct trials. The stimulus features (blue circle/red square) in both tasks, position in the distance task (above/below the reference point), distance from the reference point in the distance task, and target position (left/right) in both tasks were pseudorandomly selected. *C*, distribution of the durations of the stimuli in the duration task and the distances of the stimuli from the reference point in the distance task. *D*, penetration sites of the two monkeys. *E*, task epochs used in the reconfiguration analyses include main epochs (Pre-Go and Post-Go) and control epochs (Early Pre-Go and Pre-Go; Pre-S2off and Post-S2off). The delay (D2) between the disappearance of the second stimulus (S2off) and the go-signal was 0, 400 or 800 ms (not graphically represented). For the reconfiguration analysis, we only used trials with delays of 400 or 800 ms. Grey box ranging from Go-signal to the response indicates the reaction-movement time (RMT). *F*, heatmap of the distribution of bin/lag combinations for the pair assemblies identified by the algorithm in either task. *Bin* denotes the characteristic time scale at which the assembly was detected, whereas *lag* refers to the time latency, expressed in time bins, between the activation of the first and the second neuron in the pair assembly. Colorbar reports the number of assemblies. Assemblies with time lags greater than or equal to 10 bins between their constituent neurons were merged together. [Colour figure can be viewed at wileyonlinelibrary.com]

1 received two chambers, each with a diameter of 18 mm, whereas Monkey 2 had a single, 27 × 36 mm chamber. After surgery, the monkeys were monitored daily by full-time on-site veterinary staff who managed the use of therapies and analgesics to ensure full post-operative recovery and the general welfare of the animals.

## Histological analysis

Selected locations were targeted for electrolytic lesions (15 mA for 10 s, anodal current). Ten days later, the animal was deeply anaesthetized and perfused with formaldehyde-containing fixative through the heart. Recording sites were plotted on Nissl-stained coronal sections by comparing the recovered electrolytic lesions and the marking pins inserted during perfusion. The majority of the recordings were made from Area 8 and Area 46, as well as a limited number from Area 12 and Area 6. The entire procedure leading to death was conducted under extremely deep anaesthesia and under the supervision of the on-site veterinary staff.

## Data collection

Eye position was tracked with an infrared oculometer from Arrington Recording (Arrington Recording, Scottsdale, AZ, USA), whereas single cells were recorded using quartz-insulated platinum-iridium electrodes (0.5–1.5 MΩ at 1 kHz), positioned by a 16-electrode drive assembly from Thomas Recording (Giessen, Germany). The electrodes were arranged in a concentric array with a spacing of 518 mm. Spike data were discriminated using the Multichannel Acquisition Processor (Plexon, Dallas, TX, USA) for online discrimination using a threshold level applied on raw data to reduce the possibility of contaminating putative single-unit activity with multiunit activity. Subsequently, single-unit isolation was confirmed with the Offline Sorter (Plexon), in accordance with several quality metrics: a clear isolation of spike waveform clusters in three-dimensional principal component analysis space, minimal interspike intervals, and waveform differentiation associated with each single unit through manual inspection and curation. During recording, the head position was maintained stable with a head post. The number of neurons recorded in each recording session could change across sessions. Neurons could lose isolation during recording and were subsequently discarded.

## Cell assembly detection (CADopti)

Among the various assembly detection methods available (Quaglio et al., 2018), we employed CAD (cell assembly detection). This method, developed by Russo &

Durstewitz (2017) and subsequently optimized in its latest version as CADopti (Oettl et al., 2020), has high flexibility and the capability to detect assemblies at various temporal resolutions and with any temporal activation patterns. CADopti is an unsupervised statistical method designed to identify cell assemblies within simultaneously recorded neurons by detecting recurring multiple single-unit activity patterns in spike trains. The method consists of two main parts: a pairwise statistical test used to quantify the deviation of the joint spike distribution of two neurons from the independence hypothesis, and an agglomerative algorithm that uses this statistical test iteratively to build assemblies with an arbitrary number of elements. The algorithm tests whether specific multiple single-unit activity patterns occur more frequently than expected by chance, given the firing statistics of the constituent units.

The test accounts for non-stationarities and investigates different temporal scales of coding separately. CADopti explores a user-selected range of temporal resolutions (bin sizes) and lags in neuronal activity and returns both the activity pattern of each supra-chance assembly detected and its optimal temporal resolution. Here, we explored the following range of bins and respective maximum lags: BinSizes = [0.01, 0.015, 0.02, 0.025, 0.03, 0.04, 0.05, 0.06, 0.08, 0.1] s; MaxLags = [19, 12, 9, 7, 6, 4, 3, 2, 2, 1] bins (bins of duration 0.06 and 0.08 shared the same MaxLags of 2). The term assembly has been used to indicate a variety of forms of correlation, ranging from perfect synchronizations to sequences of activations and from milliseconds to tens or hundreds of milliseconds co-ordination precision (Russo & Durstewitz, 2017). By setting these ranges, we could detect assemblies with a characteristic temporal precision smaller than 100 ms and with a maximal lag of around 200 ms between the consecutive activation of two neurons. The algorithm aims to determine whether the joint spike count distribution of neurons is significantly distant from that obtained under the null hypothesis of independent processes. Thus, after having binned the parallel spike trains, for each pair of neurons, the algorithm counts the number of times a spike of one neuron is followed by a spike of the other after $l$ bins. The value for $l$ that maximizes the joint spike count is then selected and tested. Statistical testing is performed parametrically, allowing for a quick computation and correcting for eventual non-stationarities in the time series. After, the recursive loop begins and the algorithm adds, when possible, a new neuron to the assembly set formed in the previous step, each time considering this set as a new unit to be used in the pairwise test. The iterative algorithm stops when it is no longer possible to incorporate any new neurons into a previously formed assembly set. This process is reiterated for all binning values (temporal resolutions) specified by the user, which can be analysed independently thanks to the statistical test that corrects for non-stationarity. More

information is provided in Russo and Durstewitz (2017). This algorithm allows us to identify assemblies with arbitrary numbers of elements. In the following results, we analyse both *full-size assemblies*, which are detected by the algorithm without imposing any constraints on the number of units taking part in an assembly, and *pair assemblies*, which are all assembly subpatterns of two units identified in the dataset. Note, however, that the majority of full-size assemblies are themselves pair assemblies, as a result of the limited number of neurons recorded simultaneously in each recording session. In addition, the algorithm can return both synchronous and non-synchronous co-ordination patterns. Hence, we refer to assemblies with 0-lag in the activation of the member neurons as 'synchronous' assemblies, whereas we refer using the term 'non-synchronous' assemblies to those with a sequential pattern of activation, which is expressed in the assemblies with lag greater than zero.

## Statistical analysis

For all the statistical analysis reported in this study, $P < 0.05$ was considered statistically significant. In particular, to estimate statistical significance, we performed a two-sample $t$ test, comparing the activity of a given neuron (or assembly) both in different variable-related groups of trials or in different epochs, depending on the specific analysis performed. In the analyses where multiple comparisons were performed, we corrected all the $P$ values using the false discovery rate (FDR) correction (Benjamini & Hochberg, 1995; Trainito et al., 2019). To compare statistically significant differences in percentage, we performed a chi-squared test or, in the case of a single proportion, a binomial test. Finally, to test whether a trend of increasing proportion was significantly higher than what would be expected by chance, we used the Cochran–Armitage test for trend. We specify the statistical test used for each analysis reported in the results.

## Assembly-spikes activity and full-spikes activity

In the present study, we explore neuronal coding both in isolation and as part of an assembly. Specifically, we define a unit's full-spikes activity (FSA) as the entire neuron's firing activity, and assembly-spikes activity (ASA) as the neuron's activity during the time bins in which it contributes to an assembly. Consequently, if a neuron participates in multiple assemblies, it will have a unique FSA but multiple ASA profiles, corresponding to each assembly it contributes to. We refer to *average FSA* as the average of the FSA of the assembly's constituent neurons and to *average ASA* as the average of the ASA of the assembly's constituent neurons. For each assembly, there will be a unique average FSA and average ASA.

## Concordance of coding preference

To assign each neuron its coding preference, we compared the activity recorded in trials associated with each of the two possible dichotomies of the considered variable, blue or red for the goal colour and left or right for the response direction. The dichotomy with higher associated spiking activity was selected as the coding preference, regardless of whether the difference in activity with the other possible dichotomy was statistically significant or not. We then counted how many of the detected assemblies were composed of neurons with the same coding preference. To assess statistical significance, we performed a chi-squared test to compare the percentage of these types of assemblies obtained considering the ASA with the percentage obtained considering the FSA. We also compared each percentage with what would be expected by chance, as if those preferences had been randomly assigned. In this case, we used a binomial test to assess statistical significance. The chance level of sharing the same coding preference among assembly units is 0.5 for pair assemblies, whereas it depends on the number of assembly neurons for full-size assemblies. Given $A$ the set of all the $N$ assemblies considered, and $a_i$ the $i^{\text{th}}$ assembly of that set, we define as $|a_i|$ the number of neurons in that assembly. Then, the chance level considered for such a set of assemblies was calculated as:

$$\alpha_{level} = \frac{1}{N} \sum_{a_i \in A} 2^{1-|a_i|}$$

Note that this formula is a generalization that also applies to pair assemblies.

## Reconfiguration analysis

The aim of this analysis is to provide evidence supporting the hypothesis that the dynamic coding observed by Marcos et al. (2019) across epochs before (Pre-Go epoch) and after (Post-Go epoch) the go-signal is underpinned by a network reconfiguration (Fig. 1*E*). The Pre-Go epoch corresponds to the last 350 ms before the go-signal. As performed in Marcos et al. (2019), we excluded from the analysis trials in which D2 lasted 0 s, to include only the activity of the second delay period. The Post-Go epoch was defined as the period between 200 ms after the go-signal and the response. For comparison, we also considered two other control pairs of epochs. The first pair of epochs included the Pre-S2off and the Post-S2off, corresponding to the 350 ms before and the 350 ms after S2 was turned off. The second pair of epochs consisted of the Early Pre-Go and the Pre-Go periods of the second

delay period. The Early Pre-Go included the period from 750 to 400 ms before the go-signal, whereas the Pre-Go epoch is the same as defined above. For this analysis, we included only neurons taking part in at least two assemblies. In the first part of the analysis, we considered the whole activity, without dividing the trials according to any task-related variable. In the last part, we divided the trials according to either the goal colour or the response direction. We identify the occurrence of a reconfiguration between two epochs when a neuron participated in one assembly during the first epoch and transitioned to a different assembly in the second epoch. This process is characterized by a significant increase in the neuron's ASA associated with one assembly, coupled with a significant decrease in its ASA linked to another assembly, as the neuron moves from the first epoch to the second. We also consider partial reconfigurations, in which the change in activity occurs only in one assembly.

## Population decoding

To evaluate the amount of information carried by the two different signals, FSA and ASA, we performed a population decoding procedure (Ferrucci et al., 2022; Meyers et al., 2008, 2012; Nougaret et al., 2024). First, we selected all the neurons which did not show a significant modulation for the response direction in the window from $+200$ to $+400$ ms after the go-signal (two-sample $t$ test, $P > 0.05$, $n = 1690$) and for the goal colour chosen in the same window (two-sample $t$ test, $P > 0.05$, $n = 1841$). Next, we selected only those neurons that formed at least one assembly with any other neuron recorded simultaneously and from sessions with at least 20 trials per condition ($n = 721$) in the response direction non-selective population and in the goal colour non-selective population ($n = 847$). To each of these neurons, we associated, in the two different subpopulations, its FSA and one of its ASAs, chosen at random, and thus, respectively, its full activity and the subset of that activity associated with one of the assemblies to which it belongs. In this way, a single ASA was associated with each neuron used, preserving the one-to-one correspondence between FSA and ASA. To limit the number of assemblies associated with a neuron and the consequent random selection on ASAs, we used full-size assemblies instead of pair assemblies. As a result, the vast majority of the selected neurons belonged to only one of the assemblies (78% for the response direction and 80% for the target colour). We then performed a decoding procedure within these two subpopulations, using for each neuron only its FSA or its ASA activity. We used the Neural Decoding Toolbox (Meyers, 2013) to obtain classification accuracy between right and left trials and between red and blue goal trials. For each

neuron, data were binned in the epoch of interest (200 ms bins, from $+200$ to $+400$ ms after the go-signal), and the firing rate was normalized with a z-score transformation. Trials were labelled based on the response direction (right *vs.* left) or the goal colour (red *vs.* blue) and divided into training and test trials using a $k$-fold cross-validation procedure for the response non-selective subpopulation of 721 neurons ($k = 20$, $n = 691$) and the goal non-selective subpopulation of 847 neurons ($k = 20$, $n = 805$). The classifier was trained on the activity of $k - 1$ trials and tested on the activity of all neurons in the remaining trial. This procedure was repeated $k$ times, randomly sampling a different test trial for each neuron and the average classification accuracy was calculated. We repeated the whole procedure 1000 times to obtain a distribution of classification accuracies. We thus obtained four different distributions of classification accuracy. The same procedure was then repeated, shuffling the trial condition labels (left and right or blue and red) to obtain null distributions of classification accuracy. Significant differences between FSA and ASA classification accuracy and between FSA, ASA and their respective null distributions were evaluated by calculating the proportion of the overlapping area between the probability density functions of two distributions, using an overlapping index ($\eta$) (Pastore & Calcagnì, 2019). An $\eta$ index indicating an overlap greater than 5% was considered not significant.

## Cross-temporal decoding

To characterize the temporal coding properties according to a static or dynamic scheme, we applied a cross-temporal decoding approach, commonly employed to study such coding property (Benozzo et al., 2024; Ceccarelli et al., 2023; Di Bello et al., 2024; Mendoza-Halliday & Martinez-Trujillo, 2017; Meyers et al., 2008, 2012; Spaak et al., 2017). The decoding procedure follows the steps described above. For this analysis, we included the three epochs considered in this study, aligning the FSA activity of each cell to the go-signal for the main (from $-350$ to $+550$ ms) and control (from $-750$ to 0 ms) analysis and finally to the S2 off for a further control (from $-350$ to $+350$ ms) analysis. Thereafter, FSA activity was binned in the windows of interest, using 50 ms bins, resampled every 50 ms. The trials were then labelled according to the colour of the chosen goal (blue *vs.* red) and split by the $k$-fold cross-validation procedure to select 17 trials for training and the remaining trial for testing ($k = 18$). After the z-score normalization of the activity, under the cross-temporal decoding, the classifier is tested and trained using all possible combinations of time bins, producing as output a classification accuracy matrix where the rows and columns display the time bins

used for training and testing the classifier, respectively. Finally, the entire procedure for cross-temporal decoding was performed 50 times, randomly selecting trials for cross-validation for each run. The results were then averaged across these runs.

## Results

We analysed the activity of neurons recorded in a distance and a duration discrimination task (Fig. 1) using the CADopti algorithm (see Methods). The recordings were made in the frontal cortex (Fig. 1*D*). We selected sessions with a minimum of two recorded units, resulting in a total of 1494 neurons recorded in the distance discrimination task and 1093 neurons recorded in the duration discrimination task. Looking at the recording sessions individually, we obtained a similar average number of simultaneously recorded neurons for both tasks ($3.40 \pm 0.08$, in the distance discrimination task, and $3.38 \pm 0.09$, in the duration discrimination task). We observed that approximately half of the neurons formed at least one assembly, either a pair assembly or a full-size assembly, in both the distance discrimination task (745/1494) and the duration discrimination task (556/1093). To quantify the tendency of prefrontal units to form assemblies, we compared the number of detected pair assemblies with the number of all possible pairs of neurons recorded simultaneously. Impressively, out of the 2442 possible pairs of neurons simultaneously recorded in the distance task, 612 (25.1%) were identified as pair assemblies. Similarly, in the duration task, out of the 1699 possible pairs of neurons, 461 (27.1%) were identified as pair assemblies. In addition, we identified 411 full-size assemblies (see Methods) in the distance discrimination task and 326 assemblies in the duration discrimination task, again with the comparable average number of member neurons ($2.38 \pm 0.04$ assembly units in the distance discrimination task, and $2.34 \pm 0.04$ assembly units in the duration discrimination task). As previously described in the Methods, synchronous and non-synchronous assemblies are defined as assemblies with and without a delay in the activation of member neurons, respectively.

We then focused on both the general and task-specific aspects of assemblies. We assessed the task specificity of the detected assemblies by first identifying the assemblies separately in the two tasks and then comparing them across tasks. We analysed the 248 pair assemblies identified in at least one of the two tasks, which were formed by neurons recorded together in both tasks.

We found task specificity in 42.3% (105/248) of paired assemblies with the same pair of neurons forming an assembly in only one of the tasks. In the remaining 143 pair assemblies identified between the same neurons in both tasks, we found a perfect match of the spatiotemporal structure (i.e. the associated bin and lag) in 44.8% (64/143) of the cases examined. In the remaining assemblies (79/143), excluding synchronous ones, we found an average lag difference of $48.5 \pm 4.85$ ms. However, even when a difference in lag was found in more than 97% (67/69 non-synchronous assemblies with same units but different lag) of the cases, we observed consistency in the order of activation of the assembly members. The only exceptions were two cases where assembly directionality reversed between tasks. These results suggest that the sequence of activation of the neurons within an assembly remained hard-wired and task independent.

From this point forward, the results refer to all identified assemblies, regardless of the task in which they were recorded.

In the next analysis, using only the correct trials, we studied the coding properties of neurons with respect to goal colour and response direction during the 'Reaction and Movement Time' (RMT) epoch, which extends from the go-signal to the response period (Fig. 1*E*). We identified only assemblies with a maximum of around 200 ms latency between the consecutive activations of their constituent neurons (see Methods). Figure 1*F* shows the frequencies of bin/lag pairs assigned by the algorithm to each identified pair assembly over the entire dataset.

We will consider a cell assembly active when each of the constituent neurons emits at least one spike within a specific temporal bin at the associated lag. When the assembly pattern matches two, three or more spikes of the constituent neurons, the activity level of the assembly is considered to be two, three and so forth, respectively. By observing the neuronal activity through the lens of the assembly activation, it is thus possible to separate the spikes fired by the neuron within the assembly configuration from those discharged outside of it. We define as FSA the whole firing activity of a neuron, whereas ASA is the activity of the neuron during assembly activation bins. Furthermore, we refer to *average* FSA as the average total activity of an assembly's constituent neurons and to *average* ASA as the average ASA of the assembly's constituent neurons. This approach allows us to treat an assembly as a single entity as well as to differentiate the contribution of its constituent neurons. Therefore, we refer to a *coding assembly* for a specific task-related variable when that variable is encoded at the average ASA level. We show assemblies' activity primarily for descriptive purposes and use it as a criterion for selecting assembly subpopulations. All the analyses reported have also been reproduced using the average ASA, aiming to leave no doubt that what is observed at the level of the neurons composing an assembly reflects the collective activity of their assemblies.

## Filtering

Given the sparser nature of assembly activity compared to the activity of its constituent units, and the fact that neurons may participate in multiple assemblies, assemblies can be seen as a mechanism to selectively and transiently amplify specific moments within a unit's activity. We use the word 'filtering' to refer to the fact that the assembly activity involves only a subset of the neuron's spikes, often resulting in a clearer and more precise encoding of the task variables.

Figure 2*A* shows an assembly composed of six neurons in which the average FSA of the composing units displays no response selectivity, in contrast to the clear response coding shown by the average ASA with a preference for the left response. In parallel, although the average FSA showed consistent activity for the right response, the assembly remained almost inactive during the same trials. Moreover, this assembly was activated specifically during the RMT, peaking at the movement onset and showing very low activation outside this period, indicating

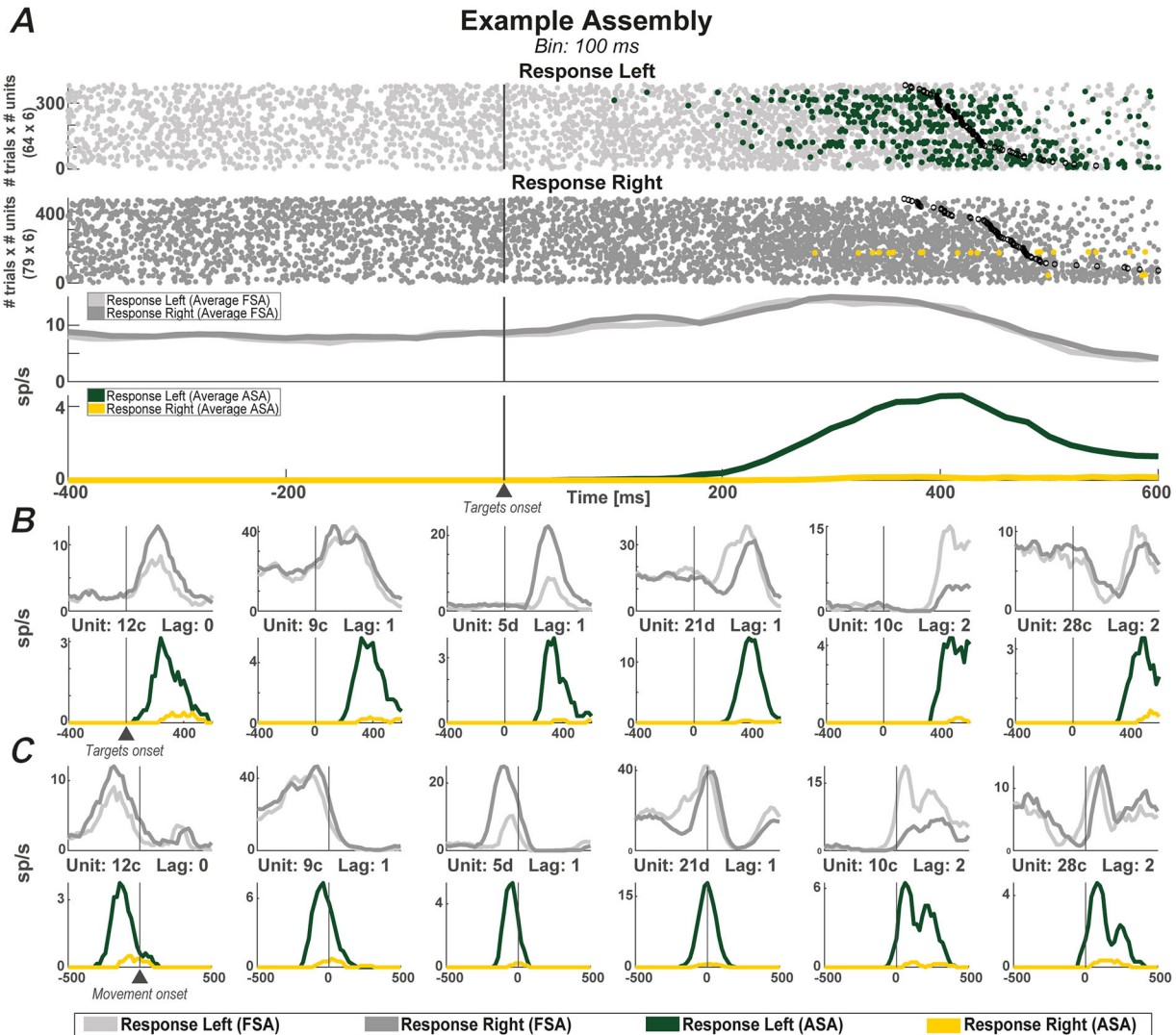

**Figure 2. Example of a cell assembly and its filtering effect on neuronal activity**
*A*, six-neuron assembly detected at a temporal resolution (binning) of 100 ms. At the top, raster plots with trials sorted by response direction are shown. Each dot represents a spike and for each trial are displayed six lines of dots, one for each neuron that belonged to the assembly. At the bottom, mean firing rates of the activity in the left and right response trials for both average FSA (light grey and dark grey) and average ASA (green and yellow). Blank black dots indicate the response time for each trial. *B* and *C*, mean firing rate as in the lower part of (*A*) but displayed for each assembly member with two different alignments for both ASA and FSA. In (*B*), the original alignment to the go-signal (target onset) is maintained, whereas, in (*C*), activities are aligned to the beginning of the movement to emphasize the shift in activation of the units of the assembly across the response time. Each cell's firing rate scale is individually adjusted to highlight its peak activity. [Colour figure can be viewed at wileyonlinelibrary.com]

that assembly specificity regarded not only the variable encoded, but also the task epoch.

Figure 2*B* and *C* show the individual activity of each of the six member neurons of the assembly aligned to the target and movement onsets, respectively. All six neurons showed a shared preference for the left response during assembly activation, as captured by their ASA. By contrast, the preference expressed by the FSA was much more heterogeneous. For example, unit 5d increased its activity for the right response at the FSA level but, when considering the ASA, its activity associated with the right response was 'filtered out'. In addition, some neurons did not show information coding in terms of FSA (9c and 28c) but did in terms of ASA. When comparing FSA and ASA, the assembly neurons showed either concordant (10c, $P < 0.01$, and 21d, $P < 0.001$, two-sample *t* test between activities in right response *vs.* left response trials) or not concordant (5d, $P < 0.001$, and 12c, $P < 0.01$, two-sample *t* test between activities in right *vs.* left trials) coding preference for response direction. Furthermore, aligning neuronal activity with the start of the movement (Fig. 2*C*) reveals that, although some neurons were specific to a particular epoch, being active solely before (5d, 9c and 12c) or after (10c) movement onset, others were active throughout both epochs (21d and 28c). This heterogeneous behaviour at the FSA level translated into a coherent coding preference and precise sequential co-ordination at the assembly level. Following the activation order identified by the assembly-specific sequence of lags, the contribution to coding for the left response at the ASA level embraced the period around the movement onset. It began with unit 12c, followed by units 9c, 5d and 21d, which fired together after one bin, to end with neurons 10c and 28c, active after a delay of two bins.

We calculated the percentage of assemblies composed of neurons with concordant coding preference in the RMT epoch with respect to the goal colour (blue or red) or the response direction (left or right). We evaluated the concordance of preference with two approaches. In the first analysis, we included only coding assemblies, which are the assemblies significantly selective for one of the two variables (Fig. 3*A* and *B*; bar plots, 'only coding assemblies') and computed how many of the constituent neurons within the assemblies shared the same coding preference. We found that the concordance of coding preferences was significantly higher than chance both when computed on the units' ASA and FSA and for both tasks (binomial test, all comparison with chance, $P < 0.001$). Given that the neurons in the assembly co-ordinate their activity, this result is not unexpected; rather, what is surprising is the extent of the effect. When both factors were taken into consideration, the percentage obtained using the ASA was significantly larger than the one obtained using the FSA (chi-squared test,

$P < 0.001$) and almost represented the entirety of the studied assemblies.

After, we extended the analysis to all the assemblies identified, regardless of their selectivity (Fig. 3*A* and *B*; bar plots, 'all assemblies'). In this case, we just added the restriction that all the assemblies included had at least one activation in the RMT epoch, that is, at least one spike of their ASA for each member neuron in the RMT epoch. We found that, in the 1050 pair assemblies considered, the concordance in the ASA coding preference was significantly higher than chance for both response direction and goal colour (858/1050 and 825/1050, respectively). However, when considering the FSA, a significant concordance was observed only for response direction (584/1050), albeit with low concordance. No significant difference from chance level was observed for the goal colour (544/1050), as shown in Fig. 3*A*. This result suggests that considering only the FSA may obscure coding-related properties of neurons that emerge only considering their co-ordinated activity within a particular assembly. We obtained comparable results on the 707 full-size assemblies with at least one activation in the RMT epoch, as shown in Fig. 3*B* (549/707 for the response direction and 526/707 for the goal colour considering the ASA; 337/707 for the response direction and 307/707 considering the FSA). A statistical significance difference from chance was found considering the ASA for both variables (bnomial test, $P < 0.001$). At the FSA level, no significance was found (goal colour, 307/707) or a barely significant result (response direction, binomial test, $P < 0.05$). Once again, the comparison between ASA and FSA was highly significant (chi-squared test, $P < 0.001$). Table 1 reports the exact *P* values.

In addition, the example assembly in Fig. 2 suggests that neurons can contribute to encode information with their ASA despite lacking selectivity with their FSA. This filtering property enables neurons, which may lack selectivity when analysed individually, to assist in transmitting information to downstream regions through co-ordination with selective neurons. Thus, this process can effectively amplify the activity of a few selective neurons, facilitating a reliable activation of downstream targets. We quantified this property by focusing on the 681 neurons that formed assemblies and that discriminated neither the response direction nor the goal colour with their FSA in the RMT epoch. As shown in Fig. 3*C*, even neurons for which the FSA was non-discriminative for any task variable could contribute to encode one or more variables when considering their ASA. As expected, the more assemblies a neuron belongs to, the more probable it is that coding properties not evident at the FSA level emerged at the assembly level. For example, when we examined the neurons involved in two assemblies during the RMT epoch, we found that 20% of them contributed to encode either response or goal variables at the ASA

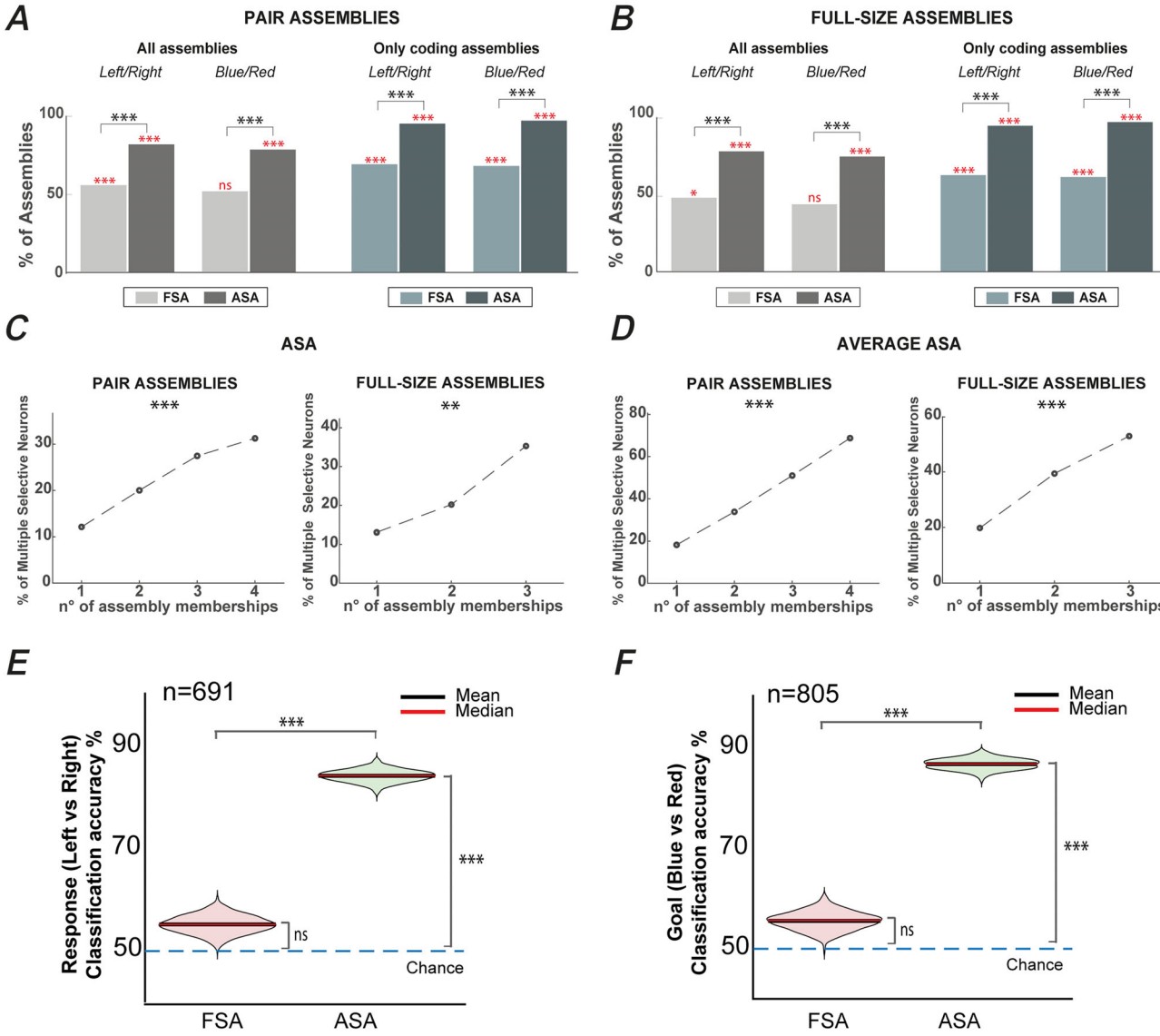

**Figure 3. Same preference and contribution to coding**

*A* and *B*, percentage of pair assemblies (light grey and dark grey plots in (*A*) and full-size assemblies (light grey and dark grey plots in (*B*) composed of neurons sharing the same coding preference. Concordance in preference was assessed for the response direction and the goal colour. The same statistics are shown for only coding assemblies, again divided by pair assemblies (light blue and dark blue plots in (*A*) and full-size assemblies (light blue and dark blue plots in (*B*). In (*A*) and (*B*), red asterisks (*) indicate statistical significance in comparison with chance level, whereas black asterisks (*) indicate statistical significance in the comparison between ASA and FSA. *C*, percentage of non-discriminative neurons at the FSA level that contributed to discriminating either the goal colour or the response direction in at least one assembly when they were considered with their ASA (two-sample *t* test, *P* < 0.05). It is important to clarify that the sum of the four data points shown in Fig. 3*C* (left) approaches ~100% merely by chance. *D*, same as (*C*), considering average ASA. Percentage of non-discriminative neurons at the FSA level, that contributed to discriminating either the goal colour or the response direction in at least one assembly when they were considered with their ASA (two-sample *t* test, *P* < 0.05). *C* and *D*, results are reported for both pair assemblies (left) and full-size assemblies (right) divided according to the number of assemblies to which a neuron belongs. These percentages increased with the number of assemblies formed by a neuron. This increase is significantly higher than what was expected by chance (Cochran–Armitage test for trend) for both pair assemblies and full-size assemblies. *P < 0.05; **P < 0.01; ***P < 0.001; ns, non-significant. Table 1 reports the exact *P* values. *E*, classification accuracy in a population of response direction non-selective neurons at the FSA level, taking into account their FSA activity and their ASA activity in one of the assemblies to which they belong. *F*, same as (*E*) considering the population of goal colour non-selective neurons at the FSA level. In (*E*) and (*F*), ***$\eta$ < 0.001, **$\eta$ < 0.01 and *$\eta$ < 0.05; ns, non-significant; *n*, number of neurons. [Colour figure can be viewed at wileyonlinelibrary.com]

**Table 1. *P* values from statistical analysis (same preference).**

### Same preference

|  | Pair assemblies — Left/Right | Pair assemblies — Blue/Red | Full-size assemblies — Left/Right | Full-size assemblies — Blue/Red |
|---|---|---|---|---|
| All assemblies | FSA $3.0 \times 10^{-4}$ — ASA $2.8 \times 10^{-101}$ — FSA vs. ASA $5.1 \times 10^{-38}$ | FSA $0.25$ — ASA $2.9 \times 10^{-81}$ — FSA vs. ASA $6.4 \times 10^{-38}$ | FSA $0.011$ — ASA $1.9 \times 10^{-79}$ — FSA vs. ASA $2.2 \times 10^{-31}$ | FSA $0.79$ — ASA $2.5 \times 10^{-65}$ — FSA vs. ASA $2.5 \times 10^{-32}$ |
| Only coding assemblies | FSA $5.9 \times 10^{-14}$ — ASA $3.3 \times 10^{-85}$ — FSA vs. ASA $3.1 \times 10^{-21}$ | FSA $3.1 \times 10^{-12}$ — ASA $4.4 \times 10^{-92}$ — FSA vs. ASA $1.9 \times 10^{-25}$ | FSA $2.5 \times 10^{-8}$ — ASA $1.2 \times 10^{-53}$ — FSA vs. ASA $6.4 \times 10^{-16}$ | FSA $5.7 \times 10^{-8}$ — ASA $8.7 \times 10^{-53}$ — FSA vs. ASA $1.0 \times 10^{-19}$ |

### Trends

| | Pair assemblies — Becoming coding | Pair assemblies — Becoming multiple coding | Full-size assemblies — Becoming coding | Full-size assemblies — Becoming multiple coding |
|---|---|---|---|---|
| ASA | $1.2 \times 10^{-4}$ | $3.7 \times 10^{-24}$ | $0.0029$ | $4.4 \times 10^{-11}$ |
| Average ASA | $8.4e \times 10^{-13}$ | $2.2 \times 10^{-35}$ | $2.3 \times 10^{-7}$ | $1.4 \times 10^{-8}$ |

### Reconfiguration

| | Pair assemblies — Pre/Post-Go vs. Pre/Post-S2off | Pair assemblies — Pre/Post-Go VS vs. EarlyPre/Pre-Go | Full-size assemblies — Pre/Post-Go VS vs. Pre/Post-S2off | Full-size assemblies — Pre/Post-Go VS vs. EarlyPre/Pre-Go |
|---|---|---|---|---|
| ASA | $1.6 \times 10^{-7}$ | $5.0 \times 10^{-8}$ | $2.8 \times 10^{-4}$ | $7.5 \times 10^{-5}$ |
| Average ASA | $4.5 \times 10^{-13}$ | $2.3 \times 10^{-15}$ | $2.3 \times 10^{-8}$ | $2.3 \times 10^{-8}$ |

### Partial reconfiguration

| | Pair assemblies — Pre/Post-Go vs. Pre/Post-S2off | Pair assemblies — Pre/Post-Go vs. EarlyPre/Pre-Go | Full-size assemblies — Pre/Post-Go vs. Pre/Post-S2off | Full-size assemblies — Pre/Post-Go vs. EarlyPre/Pre-Go |
|---|---|---|---|---|
| Selection (ASA) | $3.3 \times 10^{-5}$ | $4.1 \times 10^{-7}$ | $0.0013$ | $1.7 \times 10^{-5}$ |
| Selection (average ASA) | $3.9 \times 10^{-6}$ | $1.2 \times 10^{-9}$ | $2.6 \times 10^{-5}$ | $3.3 \times 10^{-8}$ |
| No selection (ASA) | $1.8 \times 10^{-36}$ | $6.6 \times 10^{-54}$ | $8.9 \times 10^{-21}$ | $2.3 \times 10^{-33}$ |
| No selection (average ASA) | $1.5 \times 10^{-32}$ | $1.3 \times 10^{-52}$ | $4.0 \times 10^{-20}$ | $1.5 \times 10^{-34}$ |

*P* values from the statistical analysis of the concordance in preference. The chance level was set to 0.5 when considering pair assemblies (i.e. we considered an equal probability to have the same preference against to have different preferences). In the full-size assemblies, differently from pair assemblies, the probability of having the same preference for all the neurons in each assembly is strictly related to the number of neurons in each assembly. For this reason, the chance level was calculated considering assembly numerosity (see Methods). To test the deviation from chance level, we utilized a binomial test. Additionally, we used a chi-squared test to compare FSA and ASA. Trends: *P* values of the Cochran–Armitage test for trend used to assess significance on the increasing trend of the fraction of neurons contributing to coding or multiple coding only at the assembly level. Reconfiguration—Partial Reconfiguration: *P* values of the chi-squared test of the comparison of the reconfigurations in the Pre-Go/Post-Go epochs with the two control pairs of epochs.

level in at least one assembly. If we consider neurons forming three assemblies, this percentage increased to 27% for pair assemblies and 35% for full-size assemblies, as shown in Fig. 3*C* (two-sample *t* test, FDR corrected, $P < 0.05$). Hence, it appears that the likelihood of a unit to differentiate task variables with its ASA increased with the number of assemblies it participates in. We tested this hypothesis and found a significant positive correlation between the probability of contributing to encode for a task variable and the number of assembly memberships (Cochran–Armitage test for trend) for both pair assemblies ($P < 0.001$) and full-size assemblies ($P < 0.01$). The results were comparable when considering the average ASA, as shown in Fig. 3*D*. Table 1 reports the exact *P* values.

To further assess how neurons contribute to encode information at the population level with their ASA we performed a population decoding in two different subpopulations of FSA non-selective neurons for both the response direction and the goal colour. Population decoding was performed in a window of 200 ms (from +200 to +400 ms after the go-signal). Figure 3*E* and *F* shows the classification accuracy for the right/left response (Fig. 3*E*) and for the red/blue goal (Fig. 3*F*) after the go-signal in a population of non-selective neurons when considering their FSA and when considering their ASA in one of the full-size assemblies to which they belong. However, in ∼80% of the cases, the selection is trivial because only one ASA is associated with a neuron. Classification accuracy is significantly higher for the ASA, indicating that, in a population of neurons that do not contribute to information encoding at the FSA level, coding properties can emerge when considering only a subset of their activity (ASA) co-ordinated with other neurons. A comparison of the two activities with the null distribution revealed that, as expected, for both variables analysed, whereas FSAs did not reach a level significantly above chance, ASAs did so with high statistical significance ($P < 0.001$).

## Flexibility between-variables

We investigated whether distinct encoding properties emerged from the flexibility shown by neurons when participating in different assemblies. Figure 4*A* and *B* shows the activity of an example neuron (12a) in two assemblies. Its FSA persisted during the delay and after the go-signal without displaying any selectivity, neither for the goal colour (Fig. 4*A*) nor for the response direction (Fig. 4*B*). By contrast, when considering its ASA, the neuron contributed to encode during the RMT the goal in Assembly 1 and the response in Assembly 2. This neuron did not contribute to encode in the RMT the goal in Assembly 2 ($P = 0.73$) and the response in Assembly

1 ($P = 0.48$). This example shows how neurons can contribute to the encoding of different information when active within different assemblies. We term this property as *flexibility between-variables*. This flexibility leads to the simultaneous contribution of a neuron to the encoding of multiple variables, as is the case of the neuron in Fig. 4*A* and *B*, a phenomenon we refer to as multiple selectivity. We then asked how frequently multiple selectivity emerged. We tested through an assembly-based analysis how strongly the flexibility between-variables enhanced the coding capability of the neurons. For this analysis, we counted the number of times a neuron contributed to encode multiple variables, which in our specific case means to contribute to the encoding of both the goal and the response in the RMT epoch. We did this by considering the ASA of neurons encoding none or only one of the two variables considered and quantifying how often a neuron can support the encoding of goal and response information when considering its contribution to coding within different assemblies. Figure 4*C* shows that the percentage of neurons with multiple selectivity at the ASA level increased with the number of assemblies in which the neurons participated. This percentage passed from less than 5%, for neurons taking part to only one assembly (in this case, indeed, no advantage is gained considering the ASA), to over 20%, when considering neurons taking part in three assemblies (two-sample *t* test, FDR corrected on both the number of assemblies and the number of variables considered, $P < 0.05$). This increasing trend was significantly above chance for both pair and full-size assemblies (Fig. 4*C*, Cochran–Armitage test for trend). As performed for the analysis shown in Fig. 3*C*, we repeated the analyses using the activity averaged over all neurons in a given assembly, the average ASA, again obtaining comparable results (Fig. 4*D*). Table 1 reports the exact *P* values.

## Dynamic coding reveals network reconfiguration through changes in assembly-co-ordination

In our previous work on the same dataset (Marcos et al., 2019), we showed that the encoding of task goals in the distance discrimination task was very dynamic across the goal-action transformation. This highly dynamic goal coding was expressed through the specific selectivity modalities of single neurons: either the goal was encoded exclusively before by some neurons or after the go-signal by other neurons, or the neurons switched their goal preferences between the two epochs (Marcos et al., 2019). Switch cells were defined as such for switching coding preference across the go-signal; for example, showing higher activity with their FSA for the blue goal before and for the red goal later. However, not all neurons displayed such switching activity, and some maintained a persistence in

the goal preference. Building on this previous work, we aimed to test the hypothesis that such high dynamism of coding during goal-action transformation reflects an enhanced network reorganization at the assembly level, compared to other task epochs characterized by stable coding. For this reason, the following analyses were not performed on the RMT epoch as before, but on specific pairs of epochs inherited from the work of Marcos et al. (2019) (for a detailed description of these epochs, see Methods).

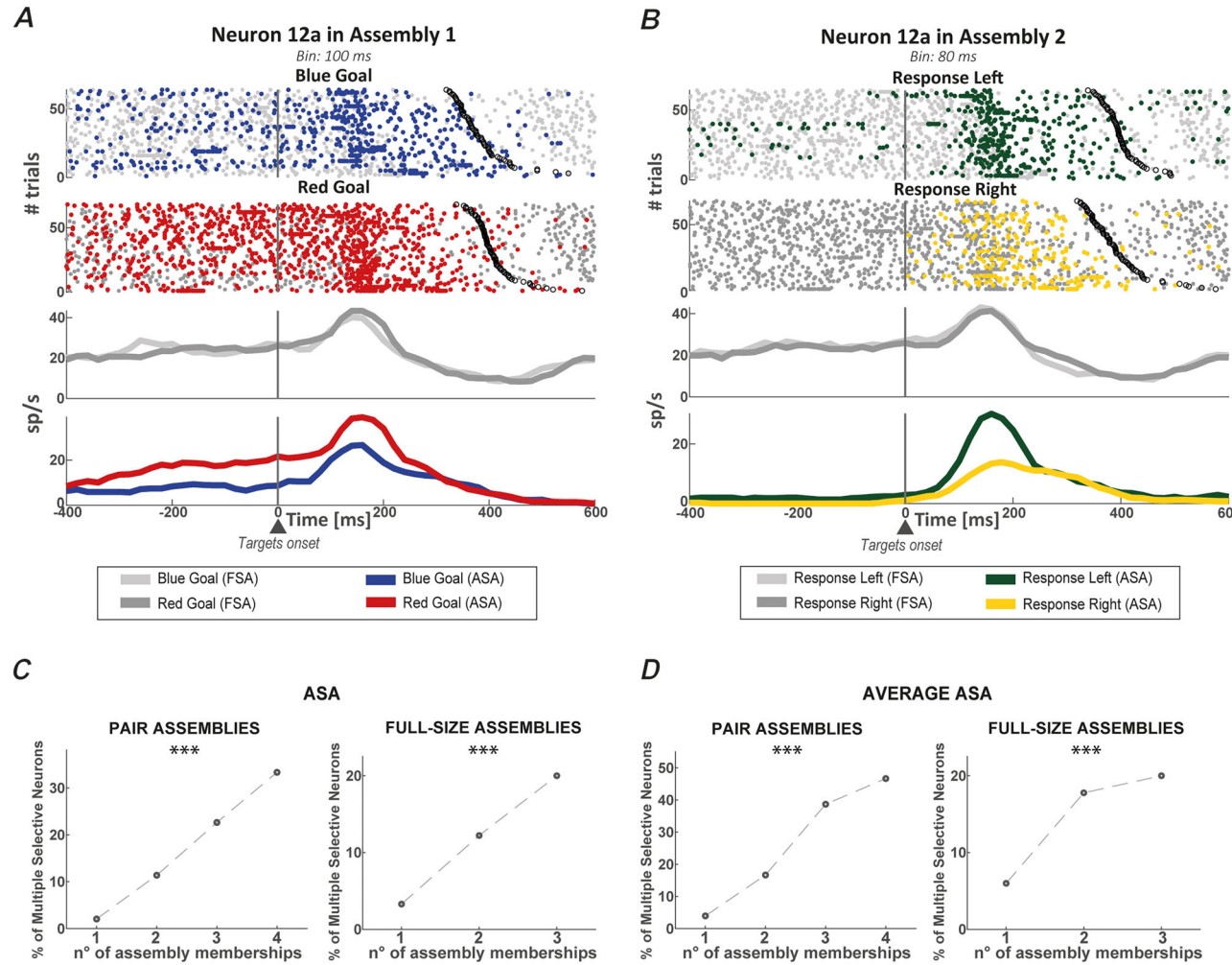

**Figure 4. Multiple selectivity of neurons taking part in multiple assemblies**
Assembly 1 is formed by neuron 12a and neuron 41a (not shown) and was detected at a temporal resolution of 100 ms. Assembly 2 is formed by neurons 12a and 9a (not shown) and was detected at a temporal resolution of 80 ms. *A*, raster plot of the activity of neuron 12a at the level of both FSA (light grey and dark grey dots) and ASA (blue and red dots) with mean firing rates at the bottom. Activities are sorted by blue and red goals. *B*, same scheme as in (*A*), but with activities sorted by left and right responses. Light grey and dark grey dots represent FSA, whereas green and yellow dots represent ASA. Neuron 12a, which was not selective for either the response or the goal, had coding properties only when its activity was considered as part of the assembly activation in the different assemblies to which it belongs. In Assembly 1 (*A*), neuron 12a contributed to encode the red goal but not the response direction (not shown). On the other hand, neuron 12a contributed to encode the left response in Assembly 2 but not the goal (not shown). *C*, percentage of non-multiple selectivity neurons, referring to neurons encoding none or only one of the two variables (response direction or goal colour) with their FSA, that contributed to the coding of both variables (multiple selectivity) with their ASA. *D*, same as (*C*) considering average ASA. Percentage of non-selective neurons at the FSA level that were part of at least one coding assembly, that is, selective for either of the two variables with its average ASA. *C* and *D*, results are reported for both pair assemblies (left) and full-size assemblies (right) divided according to the number of assemblies to which a neuron belongs. These percentages increase significantly with the number of assemblies in which a neuron participates (Cochran–Armitage test for trend) for both pair and full-size assemblies. ***$P < 0.001$. Table 1 reports the exact *P* values. [Colour figure can be viewed at wileyonlinelibrary.com]

Our approach involved analysing changes in ASA profiles between the Pre-Go and Post-Go epochs, and quantifying assemblies' reconfigurations. We identified reconfigurations in the neurons participating in more than one assembly. For comparison, we quantified reconfigurations also in two pairs of control epochs before the goal-action transformation (the Pre-S2off *vs.* Post-S2off epochs and the Early Pre-Go *vs.* Pre-Go epochs) characterized by stable goal coding (see fig. S2 in Marcos et al. (2019). Figure 1*E* shows all the epochs used in this analysis.

We define a neural reconfiguration between epochs as a significant increase in neuron activity within one assembly coupled with a significant decrease in activity within another assembly (two-sample $t$ test, $P < 0.05$). A reconfiguration, thus, implies that a neuron engages in different computations across various assemblies in different epochs. Figure 5*A* illustrates the mechanism of reconfiguration through a graphic scheme, showing a neuron (*Neu4*) shifting its participation from one to another active assembly across the goal-action transformation process.

In Fig. 5*B*–*D*, neuron 26b represents an example of this reconfiguration. This neuron decreased its ASA in Assembly 1 from the Pre-Go to the Post-Go. Across the same epochs, it increased activity in Assembly 2. By contrast, its FSA remained comparable across the two epochs. Interestingly, although the two neurons forming assemblies with neuron 26b were epoch-specific, neuron 26b bridged the information exchange between two different assemblies.

As shown in Fig. 6*A*, considering the population of neurons forming more than one pair assembly, 7.5% (39/517) of neurons showed activity reconfiguration between the Pre-Go and Post-Go epochs, which is at least about eight times higher than the percentage found in the control epochs when considering the ASA, that is, 0.77% (4/517) between the Early Pre-Go and the Pre-Go, and 0.97% (5/517) between the Pre-S2off and the Post-S2off. Comparable results were obtained for the full-size assemblies (Fig. 6*A*). These results indicate that, as predicted, the analysis of the FSA obscure information about the network dynamic, which becomes apparent when considering the ASA. However, the chance of observing reconfigurations in our study is necessarily low as a result of the limited sample of simultaneously recorded neurons. Therefore, we also quantified partial reconfigurations by identifying neurons that showed only one significant change in the ASA between epochs. For example, a neuron might decrease its activity from the Pre-Go to the Post-Go epoch within one assembly, at the same time as showing either no change in activity between the same two epochs or even no activity in any of the other assemblies identified. In this case, hypothetically, the FSA in the second epoch, which is not attributable to any

detected assemblies within the recorded set, could reflect the activation of an assembly comprising the neuron in question and other neurons that were not recorded.

Considering the population showing no reconfiguration between Pre-Go and Post-Go, the percentage of neurons with partial reconfiguration reached 83.7% (400/478) in pair assemblies, which was almost double the percentage found in the control epochs for both the categories (i.e. 35.3% (89/315) and 45.1% (124/314) in the transition between Early Pre- and Pre-Go and between Pre- and Post-S2off, respectively). Again, we found comparable results for full-size assemblies (Fig. 6*B*, 'No selection' panel). However, it is possible that ascribing all this large fraction of partial reconfiguration to switches in assembly membership (with assemblies potentially formed with unrecorded units) may have overestimated the phenomenon. For example, if a neuron is selectively active with its FSA during only one of the two epochs, changes in the ASA would simply reflect the absence of activity in the other epoch and not the neuron's participation in another unrecorded assembly. Thus, to better quantify partial reconfiguration, we restricted the analysis to neurons with consistent FSA across epochs. This was achieved by including only neurons that showed no significant difference in the FSA between the two epochs (two-sample $t$ test, $P < 0.05$). With this selection, we observed, as expected, a reduction in the percentage of neurons undergoing a partial reconfiguration. Nonetheless, this lower percentage remained approximately twice as high as that observed in the control epochs for both pair assemblies and full-size assemblies (Fig. 6*B*, 'Selection' panel). The finding that some reconfigurations also occurred in the control epochs, even if to a lesser extent than in the goal-action transformation, indicates that reconfigurations are a widespread phenomenon in the flow of information processing of the frontal cortex. Nevertheless, the reconfigurations observed between the Pre-Go and Post-Go epochs were significantly more frequent than across the control epochs, confirming that the transition between maintaining a goal in memory and using it for action selection was a moment of heightened network reconfiguration. All the analyses on reconfiguration and partial reconfiguration were also repeated using the average ASA, with comparable results for both pair and full-size assemblies (Fig. 6*C* and *D*).

We then applied a cross-temporal decoding approach to further study the interplay between the neural reconfiguration levels and population coding patterns' stability for the chosen goal across the primary and control epochs described above. For this analysis, we trained a linear classifier on FSA activity (1216 neurons taking both tasks together) from one-time bin and tested it on another time bin and vice versa. Iteratively applying this approach results in a classification accuracy matrix, where off-diagonal values represent the similarity in neural

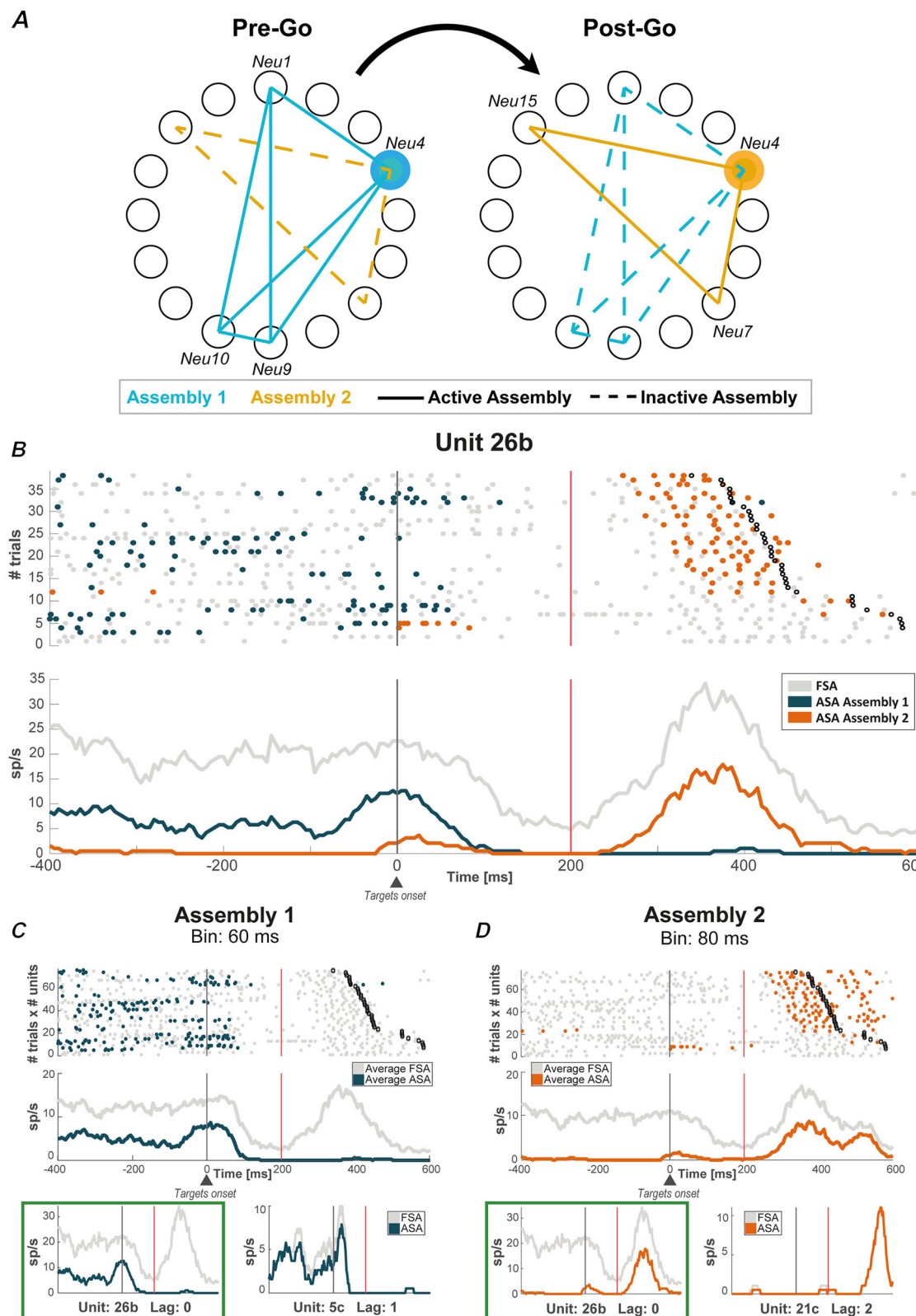

**Figure 5. Reconfiguration**
*A*, schematic representation of a reconfiguration. Filled and dotted lines indicate, respectively, active and inactive assemblies. Neuron Neu4 belongs to both Assembly 1 (cyan) and Assembly 2 (yellow). It is active with its ASA in Assembly 1 only in the Pre-Go epoch when it co-ordinates with Neu1, Neu9 and Neu10. The same neuron is active with its ASA within Assembly 2 only in the Post-Go epoch when it co-ordinates its firing rate with neurons

Neu15 and Neu7. *B*, neuron (26b) with an activity reconfiguration between the Pre- and Post-Go epochs through its participation in different assemblies, active only before (Assembly 1) or only after (Assembly 2) the go-signal. The activity is displayed without sorting by any variable. *C*, upper: the raster plot and the average FSA and ASA of the neurons of Assembly 1 are shown; lower: the individual mean ASA of the two neurons composing this assembly, one on the left (unit 26b), as shown in (*A*), and the other (5c) on the right. *D*, same as in (*C*) but for Assembly 2 with a different neuron (21c) in the assembly with neuron 26b. In (*B*) and (*D*), the black vertical bar indicates target onset (i.e. the end of the Pre-Go epoch) and the red vertical bar indicates the beginning of the Post-Go epoch, which ends at the response time (black circles). In (*C*) and (*D*), the green boxes indicate the neuron (unit 26b) shown in (*B*). [Colour figure can be viewed at wileyonlinelibrary.com]

coding patterns across time bins, indicating whether a static or dynamic coding scheme is employed over time (Spaak et al., 2017). In this context, a pronounced off-diagonal reduction in accuracy values compared to on-diagonal values (Ceccarelli et al., 2023) indicates a dynamic coding scheme. By contrast, the lack of off-diagonal reduction indicates a static coding scheme over time. When we applied this analysis in the principal epochs (Fig. 6*E*, left colour map), we observed a small or absent off-diagonal reduction in the classification accuracy throughout the Pre-Go epoch until ∼200 ms after go-signal, which suggested the occurrence of a strong static coding scheme. However, in line with previous analyses of neuronal reconfiguration, later in the Post-Go epoch, we found a marked off-diagonal reduction, which suggested a robust dynamicity of population activity patterns within and between the investigated epochs, underlying a dynamic coding scheme. Because cross-temporal decoding, unlike the preceding reconfiguration analysis, depends on neural patterns representing the animal's chosen goal, such information in the duration task is not available in the early phases of the control epochs in a fraction of trials considering that when the second stimulus is shorter the goal colour can be determined only after the stimulus is turned off. Thus, for the control epochs (Fig. 6*F*, colour maps), we have restricted the cross-temporal decoding analysis to neural populations recorded in the distance task (799 neurons), where all the trials can be used for the analysis subsequent to the goal colour in the distance task can be represented very early subsequent to the presentation of the second stimulus. Consistent with the reconfiguration analysis, we found in both the Early Pre-Go/Pre-Go (Fig. 6*F*, left colour map) and Pre-S2off/Post-S2off (Fig. 6*F*, right colour map) control epochs, an almost complete lack of off-diagonal reduction, indicative of a strong static coding scheme during these task epochs. For the sake of completeness, we also repeated the analysis for the main epoch (Fig. 6*E*, right colour map). Using only the distance task, we found comparable results to those obtained with both tasks.

In our previous analyses, we focused on determining whether the network underwent reconfiguration between the Pre-Go and Post-go epochs by changing assembly membership. Now, we shift our attention to switch cells,

as reported by Marcos et al. (2019), and ask whether the change in the information encoded between the Pre-Go and the Post-Go periods was associated with changes in assembly membership. In Fig. 7, cell 9b appears to function as a switch cell, displaying high FSA initially for the blue goal during the Pre-Go period and subsequently for the red goal in the Post-Go period (see the FSA in the upper part of the green boxes). We also observed that cell 9b changed its assembly membership from the Pre-Go to the Post-Go periods, with assembly-specific coding preference with its ASA for the blue and red goals during these epochs (see the ASA in the bottom part of the green boxes). This dual coding shows that the overall activity of neurons can be decomposed into more variable-specific components of the network in which the neuron operates.

We divided the neurons that encoded at the FSA level the goal colour in the Pre- and Post-Go epochs into persistent (non-switch) neurons, which maintained the same goal preference between epochs, and switch neurons, which instead changed goal preference. We asked whether switch neurons were special, showing a more pronounced reconfiguration than persistent neurons. Instead, we observed similar proportions of both reconfigurations and partial reconfigurations in the two categories of cells. In the persistent neurons, we observed that 10.2% (5/49) underwent activity reconfiguration. Of the remaining neurons, 41 out of 44 (93.2%) showed partial reconfiguration, whereas only three out of 44 (6.8%) did not show any reconfiguration (for an example neuron, see Fig. 8). Regarding switch neurons, we found that five out of 34 (14.7%) showed activity reconfigurations. Of the remaining neurons, 28 out of 29 (96.6%) showed a partial reconfiguration, whereas only one out of 29 (3.4%) did not show any reconfiguration at all (for an example neuron, see Fig. 7). These results show that network reconfiguration occurs regardless of the switching properties of neurons at the FSA level.

Figure 8 shows an example of reconfiguration in a persistent neuron (cell 28e, see the FSA in the upper part of the green boxes). This neuron changed its assembly participation from Assembly 1 to Assembly 2 between the Pre-Go to the Post-Go epochs. In terms of coding properties, it contributed with its ASA to encode the blue goal when active in Assembly 1 and the right response

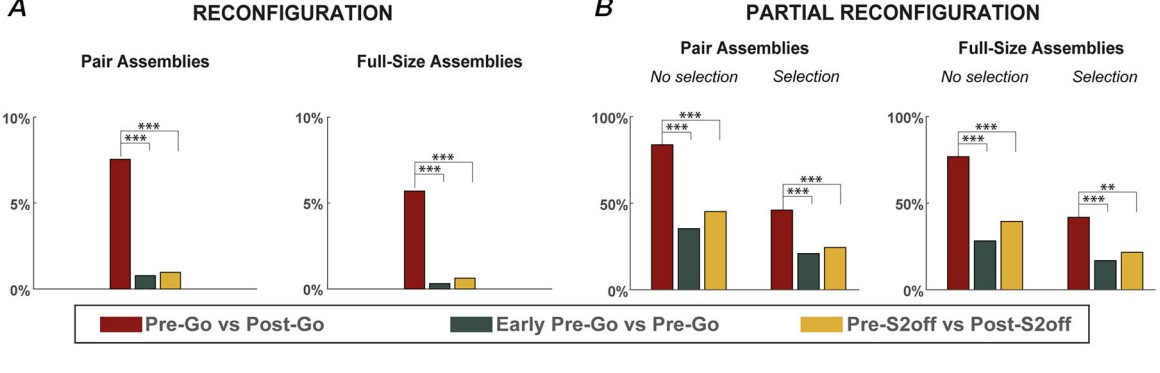

**Figure 6. Reconfiguration and cross-temporal decoding**
*A*, percentage of neurons exhibiting activity reconfigurations between each of the three pairs of epochs considered. The results are presented for both pair and full-size assemblies. *B*, same as in (*A*) but considering partial activity

reconfigurations. The results are presented both for the entire population of neurons belonging to more than one assembly and for the selected population with no statistically significant difference in the FSA between each pair of epochs and for both pair and full-size assemblies. *C*, percentage of neurons with activity reconfigurations between each of the three pairs of epochs but considering the average ASA. The same analysis was performed for both pair assemblies and full-size assemblies. *D*, same as in (*C*) but considering partial activity reconfigurations. We considered two different populations: the entire population of neurons belonging to more than one assembly (no selection panel) and the subpopulation with no statistically significant difference in the FSA between each pair of epochs (selection panel). The same analysis was performed for both pair assemblies and full-size assemblies. In (*A*) to (*D*), statistical significance was assessed using a chi-squared test. **$P < 0.01$, ***$P < 0.001$. Table 1 reports the exact *P* values. *E*, cross-temporal population decoding for the goal colour in the main epochs of interest (Pre-Go *vs.* Post-Go). In the left colour map, we used both tasks, whereas, in the right colour map, we used only the distance task as a control. *F*, cross-temporal population decoding for the goal colour in the control epochs (early Pre-Go *vs.* Pre-Go, left colour map, and Pre-S2off *vs.* Post-S2off, right colour map). In this case, only data from the distance task were used. Training and testing time bins are reported on the *y*- and *x*-axis, respectively, and the values are colour-coded in the classification accuracy matrices. [Colour figure can be viewed at wileyonlinelibrary.com]

when active in Assembly 2, respectively. Although the encoding of the blue goal could be also observed at FSA level, although with a less clear tuning than with the ASA, the response direction was encoded with a reversed preference, from left to right, when considered at the FSA or ASA level, respectively. This flexibility enabled the neuron to contribute potentially to the encoding of multiple variables by co-ordinating its activity with different groups of neurons, which can be interpreted as a signature of the transformation of the goal encoded in one assembly into the action encoded in the other.

## Discussion

Cell assembly analysis provides an alternative viewpoint on neuronal coding, moving the focus from the traditional single-neuron perspective to emphasize the dynamic role of individual neurons within collective coding mechanisms. We showed that neurons of the same assembly tend to take part in the same computation, sharing similar coding properties, and acting thus as a processing unit. Furthermore, we found that neurons in different assemblies displayed different coding preferences and timing of activation based on the specific assembly and the neurons with which they co-ordinate, which could not be captured by studying their overall activity regardless of their activation in an assembly. This flexibility contrasts with the fixed spike train of an isolated neuron when it is considered independently of its co-ordinated activity with other neurons. We discuss our findings within the context of previous studies on neural flexibility and multiplexing, as well as in the context of the goal-action transformation process, where we tested whether a strong network reconfiguration underlies a highly dynamic coding.

### Flexibility

One property of the assemblies that we have addressed is related to neural flexibility and refers to the ability of single neurons to contribute to encode different information when active in different assemblies. In this section, we will discuss the different interpretations of flexibility in neuroscience to clarify its distinctions from what we have described as assembly flexibility.

In imaging studies, flexibility can be expressed, for example, as changes in functional connectivity between tasks, between task and rest (Cohen & D'Esposito, 2016), during task execution, especially in the frontoparietal network (Cole et al., 2013) or just during rest (Yin et al., 2016).

At the population level, flexibility has been described in single-cell studies in terms of dynamic *vs.* static coding schemes, which reflect selectivity switching or transient selectivity over time (Spaak et al., 2017). Flexibility in the form of dynamic coding has been recently shown to coexist with static coding in the same brain region (Ceccarelli et al., 2023; Enel et al., 2020; Murray et al., 2017).

Flexible coding has been reported also in response to external changes in contingencies. During an extinction paradigm in rats, although the average responsiveness of the medial prefrontal cortex to task stimuli remained unchanged, changes in single-unit responsiveness co-ordinated across the population in anticipation of behavioural changes (Russo et al., 2021). Although these studies highlighted a highly flexible coding in the prefrontal cortex, they did not investigate the relationship between dynamic coding and changes in the co-ordination of neurons within assemblies.

At the assembly level, cross-correlation studies have revealed that neurons have highly dynamic synchronization patterns in response to different task events, tasks and learning phases (Nougaret & Genovesio, 2018; Sakurai & Takahashi, 2006; Tsujimoto et al., 2008). These correlations between different populations of neurons have been linked to various functions, including prospective and retrospective memories (Genovesio et al., 2005, 2006), and it has been shown that they can change flexibly between tasks (Sakurai & Takahashi, 2006).

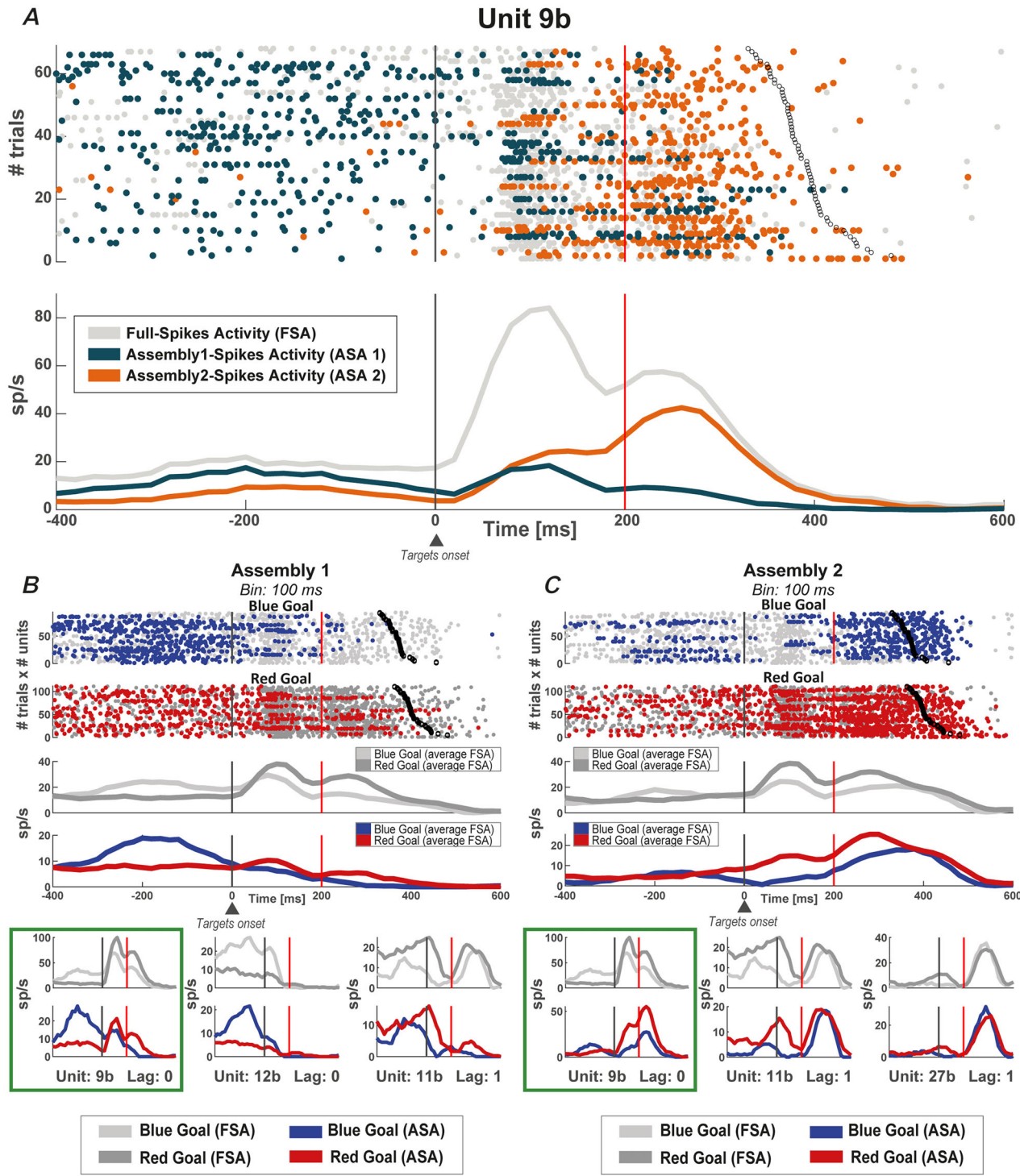

**Figure 7. Activity reconfiguration in a switch neuron**

*A*, neuron (9b) with an activity reconfiguration between Pre- and Post-go epochs through its participation in different assemblies. *B*, upper: the raster plot and the average FSA and ASA of neurons in Assembly 1 are sorted by goal colour; lower: the individual mean ASA for the two-goal colours of the three neurons composing Assembly 1 are shown: the first (unit 9b) is displayed on the left, as shown in (*A*), the second (12b) is displayed in the centre and the third (11b) is displayed on the right. *C*, same as in (*B*) but for Assembly 2 with two neurons shared with Assembly 1 (9b, 11b) and a third different neuron (27b) not shared with Assembly 1, displayed on the right. In (*A*) to (*C*), the black vertical bar indicates the target onset (i.e. the end of the Pre-Go epoch) and the red vertical bar indicates the beginning of the Post-Go epoch, which ends at the response time (black circles). In (*B*) and (*C*), the green boxes indicate the neuron (9b) in (*A*). [Colour figure can be viewed at wileyonlinelibrary.com]

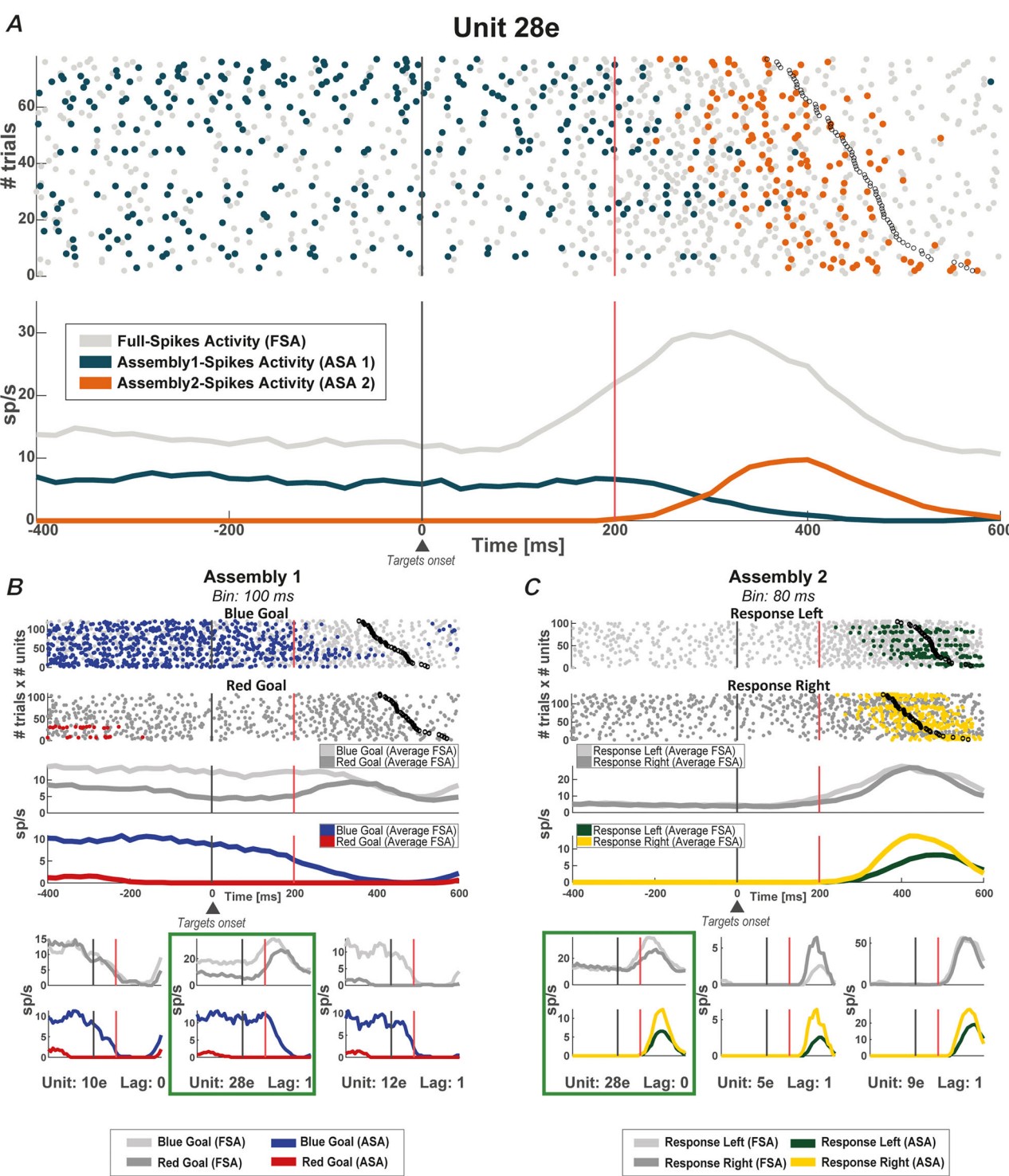

**Figure 8. Activity reconfiguration in a persistent neuron**
*A*, neuron (28e) with an activity reconfiguration between Pre- and Post-go epochs through its participation in different assemblies. The two assemblies are active in exclusive epochs. *B*, upper: the raster plot is sorted by goal colour and the average FSA and ASA of the neurons of Assembly 1 are shown sorted by each goal; lower: the individual average ASA composing Assembly 1 for the three neurons, sorted by goal colour, that contributed to encode the blue goal are presented: the first on the left (unit 10e), as shown in (*A*), the second (28e), as presented in (*A*), at the centre, and the third (12e) on the right. *C*, same as in (*B*) but for Assembly 2 with one neuron shared with Assembly 1 (28e), and with two new neurons (5e and 9e) not shared with Assembly 1. All neurons in Assembly 2 shared a preference for the right response. In (*A*) to (*C*), the black vertical bar indicates the target onset (i.e. the

end of the Pre-Go epoch) and the red vertical bar indicates the beginning of the Post-Go epoch, which ends at the response time (black circles). In (*B*) and (*C*), the green boxes indicate the neuron (28e) in (*A*). [Colour figure can be viewed at wileyonlinelibrary.com]

Although the concept of dynamic synchrony among neurons is not novel, the studies discussed before did not explicitly link assembly activity to coding and did not evaluate how the coding properties of neurons depend on the assembly to which they belong. In the present study, we have shown that at the assembly level, a neuron activity can be even more flexible, contributing to coding different variables in different assemblies. This challenges the traditional assignment of fixed coding properties to neurons based on their isolated firing. Instead, it can be useful to consider a neuron's activity within the context of the active assembly. Neurons can change the information they contribute to encode depending on the computation performed by the active assembly, which can result in different information being routed to different brain areas (Harvey et al., 2023). These results align with recent findings showing that hippocampal neurons can flexibly shift their firing phase relative to the ongoing theta rhythm of the local field potential, depending on the neuronal assemblies in which they transiently participate (Russo et al., 2024).

## Multiple selectivity

The simultaneous encoding of multiple information has been often referred to as multiplexing. It has been demonstrated that neurons are capable of multiplexing signals in the sense that they encode simultaneously multiple information (Genovesio et al., 2005; Hong et al., 2016; Kennerley et al., 2009; Ramakrishnan et al., 2017; Riehle et al., 1997). Multiplexing manifests in diverse forms, and different coding strategies have been identified for generating multiplexing. As an example of synchrony-division multiplexing, Lankarany et al. (2019) showed that stimulus intensity was encoded through asynchronous spiking, whereas high-contrast features were encoded through synchronous spiking. Time-division multiplexing is another strategy that allows for the coding of multiple stimuli by switching the coding between stimuli over time (Caruso et al., 2018; Li et al., 2016).

In the present study, we aimed to investigate the extent to which neurons' participation in different assemblies can lead to a case of extended multiplexing, which we refer to as multiple selectivity. By analysing the activity of neurons in different assemblies, we found that even non-discriminative neurons could contribute to the coding of goal and response information with the combined activity of the other assembly members. As a consequence, the participation of neurons in various assemblies, probably in a distributed manner, can increase the amount of information encoded. When examining pair assemblies and considering neurons that were part of three assemblies, we observed that more than a quarter of the initially non-discriminative neurons acquired discriminative properties when considered in co-ordination with the other assembly units. This number increased to more than a third when considering full-size assemblies. Furthermore, neurons that were initially non-multiple selective, that is, neurons encoding no or only one variable, could contribute to the encoding of multiple variables when their co-ordinate activity was considered in two or three assemblies. In other words, the activity of a neuron appears nested within the co-ordinated activity of neurons of different assemblies. In the present study, we did not attempt to relate nesting of neural activity across different assemblies to various levels of information processing hierarchy, a topic that might be better investigated in simpler organisms. Recently, Kaplan et al. (2020) provided a detailed exploration of a form of nesting in Caenorhabditis elegans, revealing a relationship between the hierarchical nesting of neuronal patterns and the multiple levels of locomotion control.

The tendency to multiple selectivity highlighted in this work is expected to become even more pronounced when tested on a larger dataset of cells, where the number of possible subsets of neurons grows exponentially and could be examined across an even greater number of assemblies. According to Buzsáki (2010), assembly activity should be viewed through the lens of downstream 'observer–reader–classifier integrator' mechanisms because 'its biological relevance can only be judged from the perspective of explicit outputs'. Consistent with this view, a neuron could then contribute to the coding of multiple types of information by participating in different assemblies, as we have observed.

Studying assemblies of neurons instead of isolated neurons can also reveal the temporal cascade of activity in the chain of active neurons and the directionality of functional connectivity between brain areas. Assemblies can encompass neurons from different areas and support the input-output communication between these areas (Domanski et al., 2023; Londei et al., 2024; Oettl et al., 2020). The flexibility of assembly formation between neurons in different brain areas during task performance has been demonstrated in a tactile discrimination task in rats by Deolindo et al. (2017). They found that assemblies between neurons in different cortical areas were more active during task execution in contrast to assemblies between neurons within the same area that were more active in the period before. In the present

study, we also observed a context dependency of assembly activation when comparing the formation of assemblies between tasks, with almost half of the neurons forming an assembly only in one task. However, we found that those neurons active in both tasks could change only their spatiotemporal structure of co-ordination but not their order of activation. These findings suggest that assembly activation is related to the network recruited by specific task computations, but the order of activation is fully hard-wired, at least when considering only assemblies of two neurons. The importance of hard-wired mechanisms in shaping the dynamics of neuronal activity has also been demonstrated at the population level by Oby et al. (2025). Using a brain–machine interface paradigm, they showed that monkeys were unable to learn to use visual feedback of their neural activity in the primary motor cortex to reverse its time course, which appeared to be rigidly determined. We can speculate that this constraint in the time course of neural activity may reflect the rigidity in the order of activation among assembly members.

The comparable level of neuronal assembly participation in the two tasks is at least compatible with the hypothesis that certain computations and aspects of neural coding might emerge from generic properties of random neural networks (Sederberg & Nemenman, 2020). Interestingly, in the network simulations of Sederberg and Nemenman (2020), neurons with correlated activity shared similar coding properties to those observed in the pair assemblies in our study. The potential importance of randomness in the organization underlying emergent properties or functions has been proposed in several domains, including spatial cognition (Mainali et al., 2025; Spalla et al., 2022), semantic memory (Zhong et al., 2025) and syntax acquisition (De Giuli, 2019). Random aspects of network organization are not incompatible with the emergence of a hard-wired temporal structure of functional connectivity, which can shape the cascade of neuronal activation. Future studies should test whether similar results can be observed when neurons are recorded in entirely different tasks or in other brain areas. More generally, working at the assembly level can help us understand the underlying mechanisms of both flexibility and rigidity in neuronal computations.

### Assembly reconfiguration and its relationship to dynamic coding during goal-action transformation

In our previous work (Marcos et al., 2019), we showed that neurons in the prefrontal cortex dynamically encoded the task's goal, through time-specific selectivities in different subpopulations and with goal preference switching, which we interpreted as the result of a reconfiguration of network activity. Such changes in the coding properties have been described as the building blocks of a dynamic coding scheme of population of neurons (Enel et al., 2020; Spaak et al., 2017). We have previously proposed that the prefrontal network that maintains information in memory passes the information to the network responsible for action planning. We hypothesized that this dynamic coding could reflect the transmission of the goal representation between neuronal assemblies, drawing an analogy with the relay race, similar to that described by Hirabayashi et al. (2013) in the perirhinal cortex in a paired association task. However, the hypothesis remained speculative until we considered the co-ordination of simultaneously recorded neurons' activity. Thus, our objective was to provide evidence that the network reconfiguration during the goal-action transformation was associated with the participation of neurons in different assemblies active at different moments during the transformation, which does not emerge at the level of the full spike activity. Considering the assembly-spike activity, we found a significantly higher proportion of neuronal reconfigurations during the most dynamic period of the goal-action transformation than in other earlier task periods characterized by stable coding of the goal information. The same result was obtained using a cross-temporal decoding analysis, which showed a transition from static to dynamic coding that was associated with an increase in assembly reconfiguration during the goal-action transformation. Reconfigurations of assemblies resemble the concept of drifting of neurons within and outside the assembly as proposed by Umbach et al. (2022) in their study of the human mesial temporal lobe. They found that assemblies with a higher degree of drifting were associated with greater memory recall performance. Taken together with our results, this suggests that the reconfiguration of assemblies can support multiple functions on different timescales: on a short timescale, processes such as goal-action transformation, and on a longer timescale, mechanisms of memory formation.

### Conclusions and future directions

Our findings suggest that taking the assembly as the unit of analysis for information coding may provide insights into how neurons orchestrate during task execution and play multiple roles in contributing to the coding of multiple variables. The moments of highly dynamic information coding observed in multiple studies (Falcone et al., 2022; Mendoza-Halliday & Martinez-Trujillo, 2017; Meyers et al., 2008, 2012; Zaksas & Pasternak, 2006) could thus express a reconfiguration of the participation of neurons in different assemblies, as in our study. Future studies should investigate in other tasks and brain areas whether the relationship between assembly reconfiguration and dynamic coding is a general phenomenon. Moving from neurons to assemblies

of neurons can also be useful to study functional connectivity. Although anatomical connectivity provides a useful starting point for understanding how different brain regions are interconnected, the analysis of the local field potentials and multiunit activity (Bardella, Franchini, et al., 2024; Bardella, Giuffrida, et al., 2024; Kunicki et al., 2019) and the co-ordination of neurons using the spiking activity can provide more information. By identifying the specific neurons forming assemblies with neurons in different brain areas and determining which information they route (Domanski et al., 2023; Harvey et al., 2023; Londei et al., 2024; Oettl et al., 2020), we will be able to study the functional interactions between brain areas in terms of the information being processed and exchanged.

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

# Additional information

## Data availability statement

Spiking data generated from the analysed dataset are available upon reasonable request from the corresponding author.

## Competing interests

The authors declare that they have no competing interests.

## Author contributions

F.L. and A.G. were responsible for the conception or design of the work. F.L., A.G., F.C., G.A., L.F., E.R. and E.B. were responsible for the acquisition, analysis or interpretation of data for the work. F.L., A.G., F.C., G.A., L.F., E.R. and E.B. were responsible for drafting the work or revising it critically for important intellectual content. All authors have approved the final version of the manuscript sunbmitted for publication and agree to be accountable for all aspects of the work. All persons designated as authors qualify for authorship, and all those who qualify for authorship are listed.

## Funding

AG has been supported by the Ministry of University and Research (MUR), project PRIN 2017 (2017KZNZLN_004). ER has been supported by #NEXTGENERATIONEU (NGEU) and funded by the Ministry of University and Research (MUR), National Recovery and Resilience Plan (NRRP), project MNESYS (PE0000006) – A multiscale integrated approach to the study of the nervous system in health and disease (DN. 1553 11.10.2022). EB has been supported by the Ministry of University and Research (MUR), project PRIN 2022 (CUP B53D23014270006).

## Acknowledgements

We thank Dr Satoshi Tsujimoto, Dr Steven P. Wise and Dr Andrew Mitz for their contributions in the experimental phase of the study.

Open access publishing facilitated by Universita degli Studi del Piemonte Orientale Amedeo Avogadro, as part of the Wiley - CRUI-CARE agreement.

## Keywords

assembly, dynamic, monkey, prefrontal, reconfiguration, static

# Supporting information

Additional supporting information can be found online in the Supporting Information section at the end of the HTML view of the article. Supporting information files available:

**Peer Review History**

