## [Peer Review History · The Journal of Physiology]

Out of the single-neuron straitjacket: neurons within assemblies change selectivity and their reconfiguration underlies dynamic coding

Fabrizio Londei, Francesco Ceccarelli, Giulia Arena, Lorenzo Ferrucci, Eleonora Russo, Emiliano Brunamonti, and Aldo Genovesio

DOI: 10.1113/JP288015

Corresponding author(s): Aldo Genovesio (aldo.genovesio@uniupo.it)

Review Timeline:

Submission Date:	31-Oct-2024
Editorial Decision:	20-Jan-2025
Revision Received:	15-May-2025
Editorial Decision:	03-Jun-2025
Revision Received:	03-Jun-2025
Accepted:	11-Jun-2025

Senior Editor: Richard Carson

Reviewing Editor: Madeleine Lowery

Transaction Report:

Dear Dr Genovesio,

Re: JP-RP-2024-288015 "Out of the single-neuron straitjacket: neurons within assemblies change selectivity and their reconfiguration underlies dynamic coding" by Fabrizio Londei, Francesco Ceccarelli, Giulia Arena, Lorenzo Ferrucci, Eleonora Russo, Emiliano Brunamonti, and Aldo Genovesio

Thank you for submitting your manuscript to The Journal of Physiology. It has been assessed by a Reviewing Editor and by 3 expert referees and we are pleased to tell you that it is potentially acceptable for publication following satisfactory major revision.

LANGUAGE EDITING AND SUPPORT FOR PUBLICATION: If you would like help with English language editing, or other article preparation support, Wiley Editing Services offers expert help, including English Language Editing, as well as translation, manuscript formatting, and figure formatting at www.wileyauthors.com/eoo/preparation. You can also find resources for Preparing Your Article for general guidance about writing and preparing your manuscript at www.wileyauthors.com/eoo/prepresources.

REVISION CHECKLIST:

We look forward to receiving your revised submission.

Yours sincerely,

Richard Carson
Senior Editor
The Journal of Physiology

REQUIRED ITEMS

- Author photo and profile. First or joint first authors are asked to provide a short biography (no more than 100 words for one author or 150 words in total for joint first authors) and a portrait photograph. These should be uploaded and clearly labelled together in a Word document with the revised version of the manuscript. See Information for Authors for further details.

- You must start the Methods section with a paragraph headed Ethical approval (https://jp.msubmit.net/cgi-bin/main.plex?form_type=display_requirements#methods).

Research must comply with The Journal's policies regarding animal experiments (<https://physoc.onlinelibrary.wiley.com/hub/animal-experiments>) and adherence to these policies must be stated in the manuscript.

Authors should confirm in their Methods section that their experiments were carried out according to the guidelines laid down by their institution's animal welfare committee, including an ethics approval reference number. The Methods section must contain a statement about access to food, water and housing, details of the anaesthetic regime: anaesthetic used, dose and route of administration, and method of killing the experimental animals.

- Your manuscript must include a complete Additional Information section, including competing interests; funding; author contributions and acknowledgements.

- Please upload separate high-quality figure files via the submission form.

- Your paper contains Supporting Information of a type that we no longer publish, including supplementary tables and figures. Any information essential to an understanding of the paper must be included as part of the main manuscript and figures. The only Supporting Information that we publish are video and audio, 3D structures, program codes and large data files. Your revised paper will be returned to you if it does not adhere to our Supporting Information Guidelines.

- Please include an Abstract Figure file, as well as the Figure Legend text within the main article file. The Abstract Figure is a piece of artwork designed to give readers an immediate understanding of the research and should summarise the main conclusions. If possible, the image should be easily 'readable' from left to right or top to bottom. It should show the physiological relevance of the manuscript so readers can assess the importance and content of its findings. Abstract Figures should not merely recapitulate other figures in the manuscript. Please try to keep the diagram as simple as possible and without superfluous information that may distract from the main conclusion(s). Abstract Figures must be provided by authors

no later than the revised manuscript stage and should be uploaded as a separate file during online submission labelled as File Type 'Abstract Figure'. Please also ensure that you include the figure legend in the main article file. All Abstract Figures should be created using BioRender. Authors should use The Journal's premium BioRender account to export high-resolution images. Details on how to use and access the premium account are included as part of this email.

EDITOR COMMENTS

Reviewing Editor:

Methods Details:

Inclusion of information on ethics and animal welfare is required.

Ethics Concerns:

The study uses data from previously published studies in macaques. No information is provided on ethical approval or animal welfare.

Comments to the Author (Required):

This study applies a previously developed assembly detection method to study the coding properties of prefrontal neurons. The reviewers agree that the approach and results are interesting and have the potential for future impact and to generate new innovative insights in the field. A number of points have been raised by the reviewers that require clarification and should be addressed before the manuscript can be accepted for publication.

As the manuscript involves data recorded in animals, the manuscript has been referred for ethics assessment. No information is provided within the manuscript on ethical approval or standards for reporting in vivo experiments. See Referee #3 comments below.

REFEREE COMMENTS

Referee #1:

The paper by Londei et al. uses an assembly-centered approach to study the coding properties of prefrontal neurons with respect to the neural assemblies to which they belong. Using a previously developed assembly detection method (Russo & Durstewitz, 2017), they extract the assembly spike activity (ASA) (i.e., the spikes associated with a particular assembly) from the full spike activity (FSA) of a unit (i.e., all the spikes generated by a neuron). This approach allows to identify neurons that may not show selectivity when considering the total spiking activity (FSA) but do show selectivity when considering only the spikes associated with a particular assembly (ASA). Consequently, it opens up new possibilities in studying a neuron's coding capacity and how the brain can implement dynamic coding of information by reconfiguring the active assemblies. A limitation of the study is the small number of simultaneously recorded neurons. As the authors note, the majority of the assemblies analyzed consisted of two-neuron assemblies. However, the approach used and the results of the paper add to our understanding of neuronal coding and will be useful to the community by paving the way to adopt a similar approach in future studies. Overall, the paper is clearly written. Below are some suggestions that I think will complement the findings of the paper.

1. The main finding of the study is that assembly-related spiking activity (ASA) reveals selectivity that is not apparent in the full spiking activity (FSA). The authors refer to this as filtering. This is shown in the paper by illustrating the activity of example neurons and by calculating how many of the neurons whose FSA was not discriminative, show selectivity when the ASA signal is considered (Figure 3 C, D). However, a more thorough comparison of the amount of information carried by the two signals is lacking. One way to explore this is to use a decoding approach that allows for a direct comparison of the amount of information carried by the FSA and ASA signals in the relevant subpopulations. The use of such an approach will strengthen the claims of the paper.

2. Another major claim of the paper is that it bridges the concept of dynamic coding with that of dynamic participation of neurons in different assemblies. However, no such direct evidence is provided. Dynamic coding can be typically studied by applying cross-temporal decoding analyses (e.g., Meyers et al., 2008, Stokes et al., 2013, Sapountzis et al., 2022). This approach will allow the authors to directly test, for example, the hypothesis that neurons showing assembly reconfiguration between epochs contribute to dynamic coding. The use of such methods may provide additional insights that will broaden the scope of the work.

3. The study employed two tasks: a distance discrimination task and a duration discrimination task. However, it is not clear from the description how the animals were instructed to perform each task. Were cues provided to indicate task type? Were the tasks presented in a blocked mode?

4. In the Results section, the authors refer to neurons as supporting "the propagation of information". Presumably, this is based on the hypothesis that coordinated firing increases the probability of influencing downstream neurons. While this is a plausible hypothesis, the study does not provide evidence that the specific neurons mentioned in the Results actually support this idea. Please either provide more information on how the study's findings support the "propagation of information" or refrain from using such phrasing when describing the findings.

5. To enhance the interpretation of Figure 3 C, D and all related figures, I would recommend changing the y-label to '% of neurons encoding a task variable' or a similar descriptive label.

6. Line 179: In the vector MaxLags= [19, 12, 9, 7, 6, 4, 3, 2, 2, 1], number 2 is written twice.

7. There appears to be some confusion regarding the different analysis epochs. To improve clarity, it would be helpful to move the schematic of the different analysis windows from Figure 5 to Figure 1, and maybe include a schematic for the Reaction and Movement Time (RMT) epoch.

Referee #2:

This manuscript reports the outcome of a straightforward idea: to apply a novel powerful procedure for the detection and quantification of statistically significant patterns of correlated activity ("cell assemblies") to a dataset of multiple simultaneous single-neuron activity recorded years ago but not yet analyzed with such methods. The results are impressive and demonstrate how much can be gained by adopting an assembly-based perspective. Of particular interest are the effects of "filtering" (to uncover assembly activity at lower firing rates than the full spiking activity of a neuron), the participation in multiple assemblies, and the flexibility and reconfiguration capability shown in the context of the specific tasks analyzed.

The report is clearly written, and obviously it relies on the previously vetted and published descriptions of the experimental methods and of the analysis procedure in the respective papers. It is not surprising, therefore, that I have no changes to recommend, and really minimal suggestions for enhanced clarity:

- The key sentence "This method consists of two main parts: a pairwise statistical test used to quantify the deviation of the joint spike distribution of two neurons from the independence hypothesis, and an agglomerative algorithm that uses this statistical test iteratively to build assemblies with an arbitrary number of elements.", which is now somewhat buried in the middle of a paragraph, could be given appropriate emphasis

- The 4 data points in Fig.3(c) sum to ca 100% just by coincidence (I would guess), which could be clarified with a short extra sentence.

- Again some extra prominence could be given to the seemingly important finding that "assembly activation is related to the network recruited by specific task computations, but the order of activation is fully hard-wired at least when considering only

assemblies of two neurons."

Moreover, I find the quantitative results very stimulating in the context of a general issue: to what extent can we take the coding revealed by these analyses to be the product of a random process? I am thinking of the near identity between assembly participation in the distance discrimination task (745/1494) and in the duration discrimination task (556/1093) - does it mean that any new task which involves neurons in these cortical areas, selected with similar criteria, will see roughly half of them participate in assemblies defined with this methodology? Similarly, in Fig.3(a) and (b), the bars for left/right and those for blue/red are nearly identical; further evidence for random coding? In a neural computation community that, until the present time, has been dominated by a functionalist perspective and an obsession with performance, random codes are beginning to bring some fresh air, from more concrete analyses of spatial cognition (Mainali, da Silveira and Burak, 2024; Spalla, Treves and Boccara, 2022) to abstract models of semantic memory (Zhong, Can, Georgiou, Shnayderman, Katkov and Tsodyks, 2024) and syntax acquisition (De Giuli, 2019), and with a couple of extra sentences in the discussion the present report can significantly contribute to this novel and most welcome perspective.

Referee #3 (ethics review):

Thank you for submitting your manuscript to The Journal of Physiology. Additional details pertaining to animal ethics and welfare are required. Please consult O'Halloran (2024): <https://physoc.onlinelibrary.wiley.com/doi/10.1113/JP286666> for an update on ethics and welfare reporting requirements for The Journal.

1. You must start the Methods section with the sub-heading "Ethical approval". Give details of the institutional ethics committee or equivalent that approved the study and include the approval number/code.
 2. You must provide detail of the source of the animals. If the monkeys were used in a previous study (or studies) please state this.
 3. Comment on housing arrangements and husbandry, particularly on the issue of single or group housing. You must clarify that animals had access to food and water.
 4. Provide details on the training regimen. The fate of the animals must be made clear. The frequency and duration of trials and the training period must be stated. Clarify if animals were restrained. Please give fulsome detail to ensure that the highest standards of welfare were adhered to in the study.
 5. You must provide detail on anaesthetic agents used to include dose and route of administration. How was an adequate depth of anaesthesia confirmed for surgical procedures? Describe surgical procedures in full. Please comment on post-operative care.
 6. Although it may appear redundant, please re-state anaesthetic dose and route and how an adequate depth of anaesthesia was determined to permit fixation. Presumably this was cardiac perfusion. Therefore, please describe the procedure. State that this was the method of euthanasia employed.
-

END OF COMMENTS

JP-RP-2024-288015 - The Journal of Physiology - Decision Letter

REQUIRED ITEMS

- Author photo and profile. First or joint first authors are asked to provide a short biography (no more than 100 words for one author or 150 words in total for joint first authors) and a portrait photograph. These should be uploaded and clearly labelled together in a Word document with the revised version of the manuscript. See Information for Authors for further details.

- You must start the Methods section with a paragraph headed Ethical approval (https://jp.msubmit.net/cgi-bin/main.plex?form_type=display_requirements#methods).

Research must comply with The Journal's policies regarding animal experiments (<https://physoc.onlinelibrary.wiley.com/hub/animal-experiments>) and adherence to these policies must be stated in the manuscript.

Authors should confirm in their Methods section that their experiments were carried out according to the guidelines laid down by their institution's animal welfare committee, including an ethics approval reference number. The Methods section must contain a statement about access to food, water and housing, details of the anaesthetic regime: anaesthetic used, dose and route of administration, and method of killing the experimental animals.

- Your manuscript must include a complete Additional Information section, including competing interests; funding; author contributions and acknowledgements.

- Please upload separate high-quality figure files via the submission form.

- Your paper contains Supporting Information of a type that we no longer publish, including supplementary tables and figures. Any information essential to an understanding of the paper must be included as part of the main manuscript and figures. The only Supporting Information that we publish are video and audio, 3D structures, program codes and large data files. Your revised paper will be returned to you if it does not adhere to our Supporting Information Guidelines.

- Please include an Abstract Figure file, as well as the Figure Legend text within the main article file. The Abstract Figure is a piece of artwork designed to give readers an immediate understanding of the research and should summarise the main conclusions. If possible, the image should be easily 'readable' from left to right or top to bottom. It should show the physiological relevance of the manuscript so readers can assess the importance and content of its findings. Abstract Figures should not merely recapitulate other figures in the manuscript. Please try to keep the diagram as simple as possible and without superfluous information that may distract from the main conclusion(s). Abstract Figures must be provided by authors no later than the revised manuscript stage and should be uploaded as a separate file during online submission labelled as File Type 'Abstract Figure'. Please also ensure that you include the figure legend in the main article file. All Abstract Figures should be created using BioRender. Authors should use The Journal's premium BioRender account to export high-resolution images. Details on how to use and access the premium account are included as part of this email.

EDITOR COMMENTS

Reviewing Editor:

Methods Details:

Inclusion of information on ethics and animal welfare is required.

Ethics Concerns:

The study uses data from previously published studies in macaques. No information is provided on ethical approval or animal welfare.

Comments to the Author (Required):

This study applies a previously developed assembly detection method to study the coding properties of prefrontal neurons. The reviewers agree that the approach and results are interesting and have the potential for future impact and to generate new innovative insights in the field. A number of points have been raised by the reviewers that require clarification and should be addressed before the manuscript can be accepted for publication.

As the manuscript involves data recorded in animals, the manuscript has been referred for ethics assessment. No information is provided within the manuscript on ethical approval or standards for reporting in vivo experiments.

See Referee #3 comments below.

REFEREE COMMENTS

We would like to thank the reviewers and the editor for the positive evaluation of our work. The constructive comments have led to a significant improvement in the manuscript. We have provided point-by-point responses to the reviewers' comments. We have included two additional analyses as requested by the reviewers. We then added two method sections to describe the newly introduced analyses. The references added in the manuscript were included in the bibliography.

As requested in the methods, we have added a section now called "Ethical Approval," and the following information: approval for animal use for the study, approval identification code, animal husbandry, access to food and water, duration of the experiment, and training scheme.

In addition, we have incorporated the supplementary figures into the main text in accordance with the journal's policy, resulting in a reorganization and renumbering of the main figures, with the exception of Figure 2, which remains completely unchanged.

During the revision, we realized that we had not specified that we had excluded neurons with zero activity at the ASA level from the trend calculation (current Figures 3C-D and 4C-D). However, we believe that it is fairer to include such neurons in the percentage calculation, since if a neuron has activity at the FSA level but not at the assembly level in the reference epoch, it should be considered as a neuron that does not acquire coding properties.

Therefore, we recalculated the trends using this methodology and obtained fully comparable results, both qualitatively and quantitatively. The significance level of the statistical tests is also the same. To show that there is no substantial effect as the result of our small change in the neurons used we report the comparison between the old and new trends and the corresponding p-values only for the reviewers in the figures below.

NEW	TRENDS			
	PAIR ASSEMBLIES		FULL-SIZE ASSEMBLIES	
	Becoming Coding	Becoming Multiple Coding	Becoming Coding	Becoming Multiple Coding
ASA	1.2e-04	3.7e-24	0.0029	4.4e-11
Average ASA	8.4e-13	2.2e-35	2.3e-07	1.4e-08

OLD	TRENDS			
	PAIR ASSEMBLIES		FULL-SIZE ASSEMBLIES	
	Becoming Coding	Becoming Multiple Coding	Becoming Coding	Becoming Multiple Coding
ASA	7.2e-05	6.7e-24	0.0026	9.4e-11
Average ASA	1.4e-12	4.5e-35	4.5e-07	1.7e-08

Referee #1

The paper by Londei et al. uses an assembly-centered approach to study the coding properties of prefrontal neurons with respect to the neural assemblies to which they belong. Using a previously developed assembly detection method (Russo & Durstewitz, 2017), they extract the assembly spike activity (ASA) (i.e., the spikes associated with a particular assembly) from the full spike activity (FSA) of a unit (i.e., all the spikes generated by a neuron). This approach allows to identify neurons that may not show selectivity when considering the total spiking activity (FSA) but do show selectivity when considering only the spikes associated with a particular assembly (ASA). Consequently, it opens up new possibilities in studying a neuron's coding capacity and how the brain can implement dynamic coding of information by reconfiguring the active assemblies. A limitation of the study is the small number of simultaneously recorded neurons. As the authors note, the majority of the assemblies analyzed consisted of two-neuron assemblies. However, the approach used and the results of the paper add to our understanding of neuronal coding and will be useful to the community by paving the way to adopt a similar approach in future studies. Overall, the paper is clearly written. Below are some suggestions that I think will complement the findings of the paper.

1. The main finding of the study is that assembly-related spiking activity (ASA) reveals selectivity that is not apparent in the full spiking activity (FSA). The authors refer to this as filtering. This is shown in the paper by illustrating the activity of example neurons and by

calculating how many of the neurons whose FSA was not discriminative, show selectivity when the ASA signal is considered (Figure 3 C, D). However, a more thorough comparison of the amount of information carried by the two signals is lacking. One way to explore this is to use a decoding approach that allows for a direct comparison of the amount of information carried by the FSA and ASA signals in the relevant subpopulations. The use of such an approach will strengthen the claims of the paper.

We thank the reviewer for his/hers time and helpful comments. We agree that a population analysis is lacking and the paper would benefit from it. In short, we asked whether higher classification accuracy could be achieved using ASA compared to FSA. We selected a two subpopulation population of non-coding neurons (two-sample t-test, $p > 0.05$) based on the response direction (right vs. left) within a 200 ms window after the go-signal (from 200 ms to 400 ms after) or on the goal color (blue vs red). From these subpopulations, we considered only neurons that formed at least one assembly, from sessions with at least 20 trials recorded per condition (691 neurons for the response direction and 805 neurons for the goal color). We then applied a decoding procedure to both subpopulations using the FSA and ASA of those neurons for comparison. In both cases, populations that were initially non-coding under FSA (i.e., showing classification accuracy at chance level) exhibited higher classification accuracy when decoding was performed using only ASA. These results were added to the manuscript by incorporating the figure below into Figure 3. We thank the reviewer for the suggestion and believe that these results significantly strengthen our claim regarding the benefits derived from assembly spike filtering.

We expanded the methods to include a new subparagraph “*Population decoding*” to explain the decoding procedure (*Methods* section): “**To evaluate the amount of information carried by the two different signals, FSA and ASA, we performed a population decoding procedure (Meyers et al., 2008, 2012; Ferrucci et al., 2022; Nougaret et al., 2024) as follows. First, we selected all the neurons which did not show a significant modulation for the response direction in the window from +200 ms to +400 ms after the go-signal (t-test, >0.05 , $n=1690$) and for the goal color chosen in the same window (t-test, $p>0.05$, $n=1841$). Next, we selected only those neurons that formed at least one assembly with any other neuron recorded simultaneously and from sessions with at least 20 trials per condition ($n= 721$) in the response direction non-selective population and in the goal color non-selective population ($n=847$). To each of these neurons, we associated, in the two different subpopulations, its FSA and one of its**

ASAs, chosen at random, and thus respectively its full activity and the subset of that activity associated with one of the assemblies to which it belongs. In this way, a single ASA was associated with each neuron used, preserving the one-to-one correspondence between FSA and ASA. To limit the number of assemblies associated with a neuron and the consequent random selection on ASAs, we used full-size assemblies instead of pair assemblies. As a result, the vast majority of the selected neurons belonged to only one of the assemblies (78% for the response direction and 80% for the target color). We then performed a decoding procedure within these two subpopulations, using for each neuron only its FSA or its ASA activity. We used the Neural Decoding Toolbox (Meyers, 2013) to obtain classification accuracy between right and left trials and between red and blue goal trials. For each neuron, data were binned in the epoch of interest (200 ms bins, from +200 ms to +400 ms after the go-signal), and firing rate was normalized with a z-score transformation. Trials were labeled based on the response direction (right vs left) or the goal color (red vs blue) and divided into training and test trials using a k-fold cross-validation procedure for the response non-selective subpopulation of 721 neurons (k=20, n=691) and the goal non-selective subpopulation of 847 neurons (k=20, n=805). The classifier was trained on the activity of k - 1 trials and tested on the activity of all neurons in the remaining trial. This procedure was repeated k times, randomly sampling a different test trial for each neuron and the average classification accuracy was calculated. We repeated the whole procedure 1000 times to obtain a distribution of classification accuracies. We thus obtained four different distributions of classification accuracy. The same procedure was then repeated, shuffling the trial condition labels (left and right or blue and red) to obtain null distributions of classification accuracy. Significant differences between FSA and ASA classification accuracy and between FSA, ASA and their respective null distributions were evaluated by calculating the proportion of the overlapping area between the probability density functions of two distributions, using an overlapping index (η) [Pastore & Calcagni 2019]. An η index indicating an overlap greater than 5% was considered not significant.”

We also added the decoding results (Results section): **”To further assess how neurons contribute to encode information at the population level with their ASA we performed a population decoding in two different subpopulations of FSA non-selective neurons for both the response direction and the goal color. Population decoding was performed in a window of 200 ms (from +200 to +400 ms after the go-signal). Fig. 3E-F shows the classification accuracy for the right/left response (Fig. 3E) and for the red/blue goal (Fig. 3F) after the go-signal in a population of non-selective neurons when considering their FSA and when considering their ASA in one of the full-size assemblies to which they belong. However, in about 80% of the cases, the selection is trivial because only one ASA is associated with a neuron. Classification accuracy is significantly higher for the ASA, indicating that in a population of neurons that do not contribute to information encoding at the FSA level, coding properties can emerge when considering only a subset of their activity (ASA) coordinated with other neurons. A comparison of the two activities with the null distribution revealed that, as expected, for both variables analyzed, while FSAs did not reach a level significantly above chance, ASAs did so with high statistical significance ($p < 0.001$)”**

2. Another major claim of the paper is that it bridges the concept of dynamic coding with that of dynamic participation of neurons in different assemblies. However, no such direct evidence is provided. Dynamic coding can be typically studied by applying cross-temporal decoding analyses (e.g., Meyers et al., 2008, Stokes et al., 2013, Sapountzis et al., 2022). This approach will allow the authors to directly test, for example, the hypothesis that neurons showing assembly reconfiguration between epochs contribute to dynamic coding. The use of such methods may provide additional insights that will broaden the scope of the work.

In accordance with the reviewer whom we thank, we implemented a cross temporal decoding analysis, considering the same epochs and neuronal populations used for the reconfiguration analysis. Through this analysis, we found, as expected, a robust relationship between task epochs with high or limited reconfiguration and dynamic and static coding schemes, respectively, in line with our hypothesis stated in the manuscript.

In the manuscript, we added in the methods a description of cross temporal decoding analysis as follows (Methods): ***“To characterize the temporal coding properties according to a static or dynamic scheme, we applied a cross-temporal decoding approach, commonly employed to study such coding property (Meyers et al., 2008, 2012; Mendoza-Halliday & Martinez-Trujillo, 2017; Spaak et al., 2017; Ceccarelli et al., 2023; Di Bello et al., 2024; Benozzo et al., 2024). The decoding procedure follows the steps described above. For this analysis, we included the three epochs considered in this study, aligning the FSA activity of each cell to the go-signal for the main (from -350 ms to 550 ms) and control (from -750 ms to 0 ms) analysis and finally to the S2 off for a further control (from -350 ms to +350 ms) analysis. Thereafter FSA activity was binned in the windows of interest, using 50 ms bins, resampled every 50 ms. The trials were then labeled according to the color of the chosen goal (blue vs red) and splitted by the k-fold cross-validation procedure to select 17 trials for training and the remaining trial for testing (k = 18). After the z-score normalization of the activity, under the cross-temporal decoding, the classifier is tested and trained using all possible combinations of time bins, producing as output a classification accuracy matrix where the rows and columns display the time bins used for training and testing the classifier, respectively. Finally, the entire procedure for cross-temporal decoding was performed 50 times, randomly selecting trials for cross-validation for each run. The results were then averaged across these runs.”***

Finally, in the Results section, we added the following: ***“We then applied a cross-temporal decoding approach to further study the interplay between the neural reconfiguration levels and population coding patterns' stability for the chosen goal across the primary and control epochs described above. For this analysis, we trained a linear classifier on FSA activity (1216 neurons taking both tasks together) from one time bin and tested it on another time bin and vice versa. Iteratively applying this approach results in a classification accuracy matrix, where off-diagonal values represent the similarity in neural coding patterns across time bins, indicating whether a static or dynamic coding scheme is employed over time (Spaak et al., 2017). In this context, a pronounced off-diagonal reduction in accuracy values compared to on-diagonal values (Ceccarelli et al., 2023) indicates a dynamic coding scheme. In contrast, the lack of off-diagonal reduction indicates a static coding scheme over time. When we applied this analysis in the principal epochs (Fig. 6E, left colormap), we observed a***

small or absent off-diagonal reduction in the classification accuracy throughout the Pre-Go epoch until approximately 200 ms after go-signal, which suggested the occurrence of a strong static coding scheme. However, in line with previous analyses of neuronal reconfiguration, later in the Post-Go epoch, we found a marked off-diagonal reduction, which suggested a robust dynamicity of population activity patterns within and between the investigated epochs, underlying a dynamic coding scheme. Since cross-temporal decoding, unlike the preceding reconfiguration analysis, depends on neural patterns representing the animal's chosen goal, such information in the duration task is not available in the early phases of the control epochs in a fraction of trials considering that when the second stimulus is shorter the goal color can be determined only after the stimulus is turned off. Thus, for the control epochs (Fig. 6F, colormaps), we have restricted the cross-temporal decoding analysis to neural populations recorded in the distance task (799 neurons), where all the trials can be used for the analysis since the goal color in the distance task can be represented very early since the period of presentation of the second stimulus. Consistent with the reconfiguration analysis, we found in both the Early Pre-Go/Pre-Go (Fig. 6F, left colormap) and Pre-S2off/Post-S2off (Fig. 6F, right colormap) control epochs, a nearly complete lack of off-diagonal reduction, indicative of a strong static coding scheme during these task epochs. For the sake of completeness, we also repeated the analysis for the main epoch (Fig. 6E, right colormap). Using only the distance task, we found comparable results to those obtained with both tasks.”

3. The study employed two tasks: a distance discrimination task and a duration discrimination task. However, it is not clear from the description how the animals were instructed to perform each task. Were cues provided to indicate task type? Were the tasks presented in a blocked mode?

We agree with the reviewer that further clarification is necessary. The tasks were presented in a block design, so no cues were necessary nor provided during the tasks to indicate task type. In addition, the appearance of the stimuli at the screen center was unique to the temporal task as was unique to the distance task the placement of the two stimuli above or below the center.

To address this concern, we have included the following sentence in the text to ensure clarity (*Methods* section): ***“The tasks were presented in a block design, so no cues were necessary nor provided during the tasks to indicate task type.”***

4. In the Results section, the authors refer to neurons as supporting "the propagation of information". Presumably, this is based on the hypothesis that coordinated firing increases the probability of influencing downstream neurons. While this is a plausible hypothesis, the study does not provide evidence that the specific neurons mentioned in the Results actually support this idea. Please either provide more information on how the study's findings support the "propagation of information" or refrain from using such phrasing when describing the findings.

The reviewer raised an important point. We indeed referred to the hypothesis that coordinated firing increases the probability of influencing downstream neurons, but it would have been more accurate to write “could support propagation of information” or to frame the

description more strictly in terms of coding. We chose to use the term "coding" to make our explanation less ambiguous.

Specifically, we have changed the following sentences:

1. In the next analysis, using only the correct trials, we studied the coding properties of neurons **in propagating information about** goal color and response direction during the "Reaction and Movement Time" (RMT) epoch, which extends from the go-signal to the response period.
→ In the next analysis, using only the correct trials, we studied the coding properties of neurons **with respect to** goal color and response direction during the "Reaction and Movement Time" (RMT) epoch, which extends from the go-signal to the response period. (*Results* section)
2. This example shows how neurons can contribute to **the propagation of different information** when active within different assemblies.
→ This example shows how neurons can contribute to **the encoding of different information** when active within different assemblies. (*Results* section)
3. We did this by considering the ASA of neurons encoding none or only one of the two variables considered and quantifying how often a neuron can support **the propagation of goal and response information** when considering its contribution to coding within different assemblies.
→ We did this by considering the ASA of neurons encoding none or only one of the two variables considered and quantifying how often a neuron can support **the encoding of goal and response information** when considering its contribution to coding within different assemblies. (*Results* section)

5. To enhance the interpretation of Figure 3 C, D and all related figures, I would recommend changing the y-label to '% of neurons encoding a task variable' or a similar descriptive label. We modified the y-label in Figure 3C and 3D reporting "% of Selective Neurons" and the y-label in Figure 4C and 4D reporting "% of Multiple Selective Neurons".

6. Line 179: In the vector `MaxLags= [19, 12, 9, 7, 6, 4, 3, 2, 2, 1]`, number 2 is written twice. We appreciate the feedback and believe that a clarification within the text is necessary. The two input vectors, "MaxLags" and "BinSizes," possess an equal number of elements, specifically 11, due to their one-to-one relationship. Indeed, each element in the "BinSizes" vector is mapped to a corresponding element in the "MaxLags" vector, indicating the maximum lag at which the coordination in the spike trains will be evaluated for that specific bin size. In the specific case of the number 2 written twice, this indicates that a maximum lag of 2 is associated with both the bin sizes of 60 ms and 80 ms. The objective was to identify the multiple of the bins that, after multiplication with the associated bin size, yields a value closer to 200 ms. In the specific case of the bins of 80 and 60 ms, this multiple was found to be the same. The calculation to be executed is as follows: $\text{BinSize} * (\text{MaxLag} + 1)$. To elucidate this concept, the following modification is made to the original sentence (Methods section): "*Here, we explored the following range of bins and respective maximum lags: BinSizes= [0.01, 0.015, 0.02, 0.025, 0.03, 0.04, 0.05, 0.06, 0.08, 0.1] sec; MaxLags= [19, 12, 9, 7, 6, 4, 3, 2, 2, 1] bins (bins of duration 0.06 and 0.08 shared the same MaxLags of 2).*"

7. There appears to be some confusion regarding the different analysis epochs. To improve clarity, it would be helpful to move the schematic of the different analysis windows from Figure 5 to Figure 1, and maybe include a schematic for the Reaction and Movement Time (RMT) epoch.

We agree with the reviewer's suggestion. We moved the schematic of the time windows into Figure 1E and we added a panel to highlight the RMT epoch.

Referee #2

This manuscript reports the outcome of a straightforward idea: to apply a novel powerful procedure for the detection and quantification of statistically significant patterns of correlated activity ("cell assemblies") to a dataset of multiple simultaneous single-neuron activity recorded years ago but not yet analyzed with such methods. The results are impressive and demonstrate how much can be gained by adopting an assembly-based perspective. Of particular interest are the effects of "filtering" (to uncover assembly activity at lower firing rates than the full spiking activity of a neuron), the participation in multiple assemblies, and the flexibility and reconfiguration capability shown in the context of the specific tasks analyzed.

The report is clearly written, and obviously it relies on the previously vetted and published descriptions of the experimental methods and of the analysis procedure in the respective papers. It is not surprising, therefore, that I have no changes to recommend, and really minimal suggestions for enhanced clarity:

- The key sentence "This method consists of two main parts: a pairwise statistical test used to quantify the deviation of the joint spike distribution of two neurons from the independence hypothesis, and an agglomerative algorithm that uses this statistical test iteratively to build assemblies with an arbitrary number of elements.", which is now somewhat buried in the middle of a paragraph, could be given appropriate emphasis

We thank the reviewer for pointing this out. We have moved the key sentence to the beginning of the paragraph to give it more emphasis.

- The 4 data points in Fig.3(c) sum to ca 100% just by coincidence (I would guess), which could be clarified with a short extra sentence.

Yes, the fact that the sum of the data points is approximately 100 is a coincidence. We clarified that by adding a short sentence (*Results* section): **"It is important to clarify that the sum of the four data points shown in the left panel of Figure 3C approaches approximately 100% merely by chance."**

- Again some extra prominence could be given to the seemingly important finding that "assembly activation is related to the network recruited by specific task computations, but the order of activation is fully hard-wired at least when considering only assemblies of two neurons."

Thank you for pointing out that this topic deserves some more attention.

We have added a paragraph that put the result in a broader context (*Discussion* section): "The importance of hard-wired mechanisms in shaping the dynamics of neuronal activity has also been demonstrated at the population level by Obi et al. (2025). Using a brain-machine

interface paradigm, they showed that monkeys were unable to learn to use visual feedback of their neural activity in the primary motor cortex to reverse its time course, which appeared to be rigidly determined. We can speculate that this constraint in the time course of neural activity may reflect the rigidity in the order of activation among assembly members. This rigidity should not be viewed as the result of an underlying set of complex connectivity rules but could also emerge from a randomly connected network of neurons.”

Moreover, I find the quantitative results very stimulating in the context of a general issue: to what extent can we take the coding revealed by these analyses to be the product of a random process? I am thinking of the near identity between assembly participation in the distance discrimination task (745/1494) and in the duration discrimination task (556/1093) - does it mean that any new task which involves neurons in these cortical areas, selected with similar criteria, will see roughly half of them participate in assemblies defined with this methodology? Similarly, in Fig.3(a) and (b), the bars for left/right and those for blue/red are nearly identical; further evidence for random coding? In a neural computation community that, until the present time, has been dominated by a functionalist perspective and an obsession with performance, random codes are beginning to bring some fresh air, from more concrete analyses of spatial cognition (Mainali, da Silveira and Burak, 2024; Spalla, Treves and Boccara, 2022) to abstract models of semantic memory (Zhong, Can, Georgiou, Shnayderman, Katkov and Tsodyks, 2024) and syntax acquisition (De Giuli, 2019), and with a couple of extra sentences in the discussion the present report can significantly contribute to this novel and most welcome perspective.

Thank you for the suggestion, accordingly to broaden the discussion of the paper in this direction we have added a sentence to put our results in the context of this hypothesis (*Discussion* section):“The comparable level of neuronal assembly participation in the two tasks is at least compatible with the hypothesis that certain computations and aspects of neural coding might emerge from generic properties of random neural networks (Sederberg et al., 2020). Interestingly, in the network simulations of Sederberg et al. (2020) neurons with correlated activity shared similar coding properties to those observed in the pair assemblies in our study. The potential importance of randomness in the organization underlying emergent properties or functions has been proposed in several domains, including spatial cognition (Mainali et al., 2025; Spalla et al., 2022), semantic memory (Zhong et al., 2024), and syntax acquisition (De Giuli, 2019). Random aspects of network organization are not incompatible with the emergence of a hard-wired temporal structure of functional connectivity, which can shape the cascade of neuronal activation. Future studies should test whether similar results can be observed when neurons are recorded in entirely different tasks or in other brain areas. More generally, working at the assembly level can help us understand the underlying mechanisms of both flexibility and rigidity in neuronal computations.”

Referee #3 (ethics review)

Thank you for submitting your manuscript to The Journal of Physiology. Additional details pertaining to animal ethics and welfare are required. Please consult O'Halloran (2024):

<https://physoc.onlinelibrary.wiley.com/doi/10.1113/JP286666> for an update on ethics and welfare reporting requirements for The Journal.

1. You must start the Methods section with the sub-heading "Ethical approval". Give details of the institutional ethics committee or equivalent that approved the study and include the approval number/code.

We added the sub-heading "Ethical Approval" at the beginning of the Methods section:

"All animal surgical and experimental procedures were approved in advance by the National Institute of Mental Health Animal Care and Use Committee [LSN_03_05] and followed the National Institutes of Health Guide for the Care and Use of Laboratory Animals (1996) and were approved by the National Institute of Mental Health Animal Care and Use Committee. The authors understand the ethical principles under which The Journal of Physiology operates and confirm that this work complies with its animal ethics checklist."

2. You must provide detail of the source of the animals. If the monkeys were used in a previous study (or studies) please state this.

Unfortunately, we do not have access to information about the origin of the animals, since previous articles that used the same task and the same animals do not report this information. We reported the articles that were published in the past using the same task/monkeys in the *Animals* subsection, *Methods* section: ***"The information about the care and welfare of the animals is as reported in previously published articles that used the same task/animals (Genovesio et al. 2009, 2011, 2012, 2015; Marcos et al. 2017; Benozzo et al. 2021, Benozzo et al. 2023)"***

3. Comment on housing arrangements and husbandry, particularly on the issue of single or group housing. You must clarify that animals had access to food and water.

We reported the information about husbandry and monkey's diet in the *Animals* subsection, *Methods* section: ***"To motivate the animals during training and neural recordings, water intake was controlled. Monkeys had full access to dry food. After the daily experimental sessions, monkeys received fresh food such as fruit and vegetables. Weight was monitored several times a week and kept above 85% of the weight of the monkey before starting the water control schedule. The animals' weight and general health conditions were carefully monitored by full-time on site veterinary staff. The monkeys were paired housed unless adverse outcomes precluded temporarily the pairing"***

4. Provide details on the training regimen. The fate of the animals must be made clear. The frequency and duration of trials and the training period must be stated. Clarify if animals were restrained. Please give fulsome detail to ensure that the highest standards of welfare were adhered to in the study.

We reported the information about training regime in the *Animals* subsection, *Methods* section: ***"The monkeys were trained prior to surgery and the start of recordings for a period of approximately 2 years, on a monday to friday schedule with two resting days during the weekend."***

We have also added information on the restraining of the head in the data collection section: ***"During recording the head position was maintained stable with a head post."***

5. You must provide detail on anaesthetic agents used to include dose and route of administration. How was an adequate depth of anaesthesia confirmed for surgical procedures? Describe surgical procedures in full. Please comment on post-operative care.

Unfortunately, we do not have access to veterinary treatment protocols, since previous articles that used the same task and the same animals did not report this information. We added this statement in the *Surgery* subsection, *Methods* section: ***“After surgery, the monkeys were monitored daily by full-time on site veterinary staff who managed the use of therapies and analgesics to ensure full post-operative recovery and the general welfare of the animals.”***

6. Although it may appear redundant, please re-state anaesthetic dose and route and how an adequate depth of anaesthesia was determined to permit fixation. Presumably this was cardiac perfusion. Therefore, please describe the procedure. State that this was the method of euthanasia employed.

Unfortunately, we do not have access to veterinary procedures during euthanasia, since previous articles that used the same task and the same animals do not report this information. We added this statement in the *Histological Analysis* subsection, *Methods* section: ***“The entire euthanasia procedure was conducted under extremely deep anaesthesia and under the supervision of the on-site veterinary staff”***

Dear Dr Genovesio,

Re: JP-RP-2025-288015R1 "**Out of the single-neuron straitjacket: neurons within assemblies change selectivity and their reconfiguration underlies dynamic coding**" by Fabrizio Londei, Francesco Ceccarelli, Giulia Arena, Lorenzo Ferrucci, Eleonora Russo, Emiliano Brunamonti, and Aldo Genovesio

Thank you for submitting your manuscript to The Journal of Physiology. It has been assessed by a Reviewing Editor and by 3 expert referees and we are pleased to tell you that it is acceptable for publication following satisfactory revision.

REVISION CHECKLIST:

We look forward to receiving your revised submission.

Yours sincerely,

Richard Carson
Senior Editor
The Journal of Physiology

REQUIRED ITEMS

- You must start the Methods section with a paragraph headed Ethical approval (https://jp.msubmit.net/cgi-bin/main.plex?form_type=display_requirements#methods).

Research must comply with The Journal's policies regarding animal experiments (<https://physoc.onlinelibrary.wiley.com/hub/animal-experiments>) and adherence to these policies must be stated in the manuscript.

Authors should confirm in their Methods section that their experiments were carried out according to the guidelines laid down by their institution's animal welfare committee, including an ethics approval reference number. The Methods section must contain a statement about access to food, water and housing, details of the anaesthetic regime: anaesthetic used, dose and route of administration, and method of killing the experimental animals.

EDITOR COMMENTS

Reviewing Editor:

The authors have satisfactorily addressed the scientific questions raised by the reviewers. As indicated by the Animal Ethics Editor (Ref #3 below), some further clarification is required to make it clear that this is a data analysis study based on previous experimental recordings. Otherwise further details of the experimental methods and adherence to the ethical policy and reporting standards would be required.

REFEREE COMMENTS

Referee #1:

All my previous comments have been addressed by the authors in a comprehensive and satisfactory way.

Referee #2:

The revision has addressed all points raised by the reviewers, in my opinion.

Referee #3 (ethics review):

Thank you for making revisions to the text.

It is unclear whether this study is exclusively a data analysis study based on experimental recordings made previously in cited published works or a de novo experimental study. Your comments in your rebuttal that some information regarding procedures and welfare are unavailable to you suggest that it is the former. Please clarify and amend the manuscript text as required.

If the study is a data analysis study then this should be stated at the outset of the Methods for clarity.

END OF COMMENTS

JP-RP-2024-288015 - The Journal of Physiology - Decision Letter

It is unclear whether this study is exclusively a data analysis study based on experimental recordings made previously in cited published works or a de novo experimental study. Your comments in your rebuttal that some information regarding procedures and welfare are unavailable to you suggest that it is the former. Please clarify and amend the manuscript text as required.

Answer:

We thank the reviewer for asking to make it more explicit that this is a data analysis paper. I confirm that the manuscript is a data analysis study based on experimental recordings made previously in cited published works.

We wrote at the beginning of the ethical section the following sentence: "The current study is a data analysis study based on experimental recordings made previously in cited published works."

Dear Professor Genovesio,

Re: JP-RP-2025-288015R2 "**Out of the single-neuron straitjacket: neurons within assemblies change selectivity and their reconfiguration underlies dynamic coding**" by Fabrizio Londei, Francesco Ceccarelli, Giulia Arena, Lorenzo Ferrucci, Eleonora Russo, Emiliano Brunamonti, and Aldo Genovesio

We are pleased to tell you that your paper has been accepted for publication in The Journal of Physiology.

Yours sincerely,

Richard Carson
Senior Editor
The Journal of Physiology

If you would like to receive our 'Research Roundup', a monthly newsletter highlighting the cutting-edge research published in The Physiological Society's family of journals (The Journal of Physiology, Experimental Physiology, Physiological Reports, The Journal of Nutritional Physiology and The Journal of Precision Medicine: Health and Disease), please click this link, fill in your name and email address and select 'Research Roundup':
<https://www.physoc.org/journals-and-media/membernews>

- **TRANSPARENT PEER REVIEW POLICY:** To improve the transparency of its peer review process, The Journal of Physiology publishes online as supporting information the peer review history of all articles accepted for publication. Readers will have access to decision letters, including Editors' comments and referee reports, for each version of the manuscript as well as any author responses to peer review comments. Referees can decide whether or not they wish to be named on the peer review history document.
- You can help your research get the attention it deserves! Check out Wiley's free Promotion Guide for best-practice recommendations for promoting your work at: www.wileyauthors.com/eoo/guide. You can learn more about Wiley Editing Services which offers professional video, design, and writing services to create shareable video abstracts, infographics, conference posters, lay summaries, and research news stories for your research at: www.wileyauthors.com/eoo/promotion.
- **IMPORTANT NOTICE ABOUT OPEN ACCESS:** To assist authors whose funding agencies mandate public access to published research findings sooner than 12 months after publication, The Journal of Physiology allows authors to pay an Open Access (OA) fee to have their papers made freely available immediately on publication.

REFeree COMMENTS

Referee #3:

Thank you for this revision. There are no further issues.